**communications** engineering

# Pillar arrays as tunable interfacial barriers for microphysiological systems
Ishan Goswami [1,2], Yongdeok Kim[1,2,4], Gabriel Neiman [1], Brian Siemons[1], Jazmin I. Velazquez[1], Kerem Yazgan [1], Tammy Ng [2], Sudipta Ashe[3] & Kevin E. Healy[1,2] ✉

We report on the design and fabrication of a circular pillar array as an interfacial barrier for microfluidic microphysiological systems (MPS). Traditional barrier interfaces, such as porous membranes and microchannel arrays, present limitations due to inconsistent pore size, complex fabrication and device assembly, and a lack of tunability using a scalable design. Our pillar array overcomes these limitations by providing precise control over pore size, porosity, and hydraulic resistance through simple modifications of pillar dimensions. Serving as an interface between microfluidic compartments, it facilitates cell aggregation for tissue formation and acts as a tunable diffusion barrier that mimics diffusion in vivo. We demonstrate the utility of barrier design to engineer physiologically relevant cardiac microtissues and a heterotypic model with vasculature within the device. The tunable properties offer significant potential for drug screening/testing and disease modeling, enabling comparisons of drug permeability and cell migration in MPS tissue with or without vasculature.

Microfluidics has been widely adopted to create microphysiological systems (MPS) that can be used to culture cells/tissues in near-physiological conditions via the provision of dynamic flow conditions, physiological ratios of tissue and culture medium, and the ability to create 3D tissue geometries. We and others have shown the application of these devices for drug testing and disease modeling[1–8]. Many of these microfluidic devices consist of a tissue chamber separated from the media channels via a diffusion barrier that controls the transport rate of biomolecules to and from the tissues. Two widely used form factors of engineered interfaces as diffusion barriers are porous membranes[9–11] and microchannel barriers[12–16].

Thin porous polymeric membranes have been used to separate the tissue chamber from the media channels[9–11]. The porous membrane allows the diffusion of nutrients and molecules, as well as allowing heterotypic cell–cell communication (i.e., communication between different types of cells in a tissue) via the incorporation of two different cell types on either side of the membrane. The advantage of this approach is the commercial availability of porous membranes made of polymers such as polyethylene terephthalate and polycarbonate. The porosity in these polymeric membranes is achieved via a track-etching process involving bombardment of the polymer with heavy ions followed by chemical etching[17]. The fabrication process is scalable and allows for wide adoption. However, there is huge variation in pore sizes and pore position in these membranes due to the stochastic distribution of pores that overlap forming unwanted larger openings, inter-pore spacing, and pore densities (Supplementary

Fig. 1A)[18,19]. Alternatively, others have created membranes out of replica molded polydimethylsiloxane (PDMS), electropsun membranes, and silicon nitride films[11,20–22]. The process of incorporating these membranes into an MPS involves functionalizing membranes/adhesive application and cumbersome alignment with the microfluidic chambers during assembly[1,2]. Furthermore, membrane integrity and porosity may be compromised during solvent cleaning and handling during assembly[18].

Another form-factor uses an array of microchannels as a diffusion barrier separating the media channels and the tissue chamber[12–16]. The hydraulic diameter of the microchannels provides the equivalent pore size achieved via polymeric membranes. These microchannel-based barriers are incorporated into the design of the master mold used to create the polymeric MPS device and therefore minimize any cumbersome alignment and additional functionalization during assembly. Furthermore, the pore size and position of the microchannels are easily controlled during the design and fabrication of the master mold. However, patterning of these microchannel arrays, including pitch and length of the channels, is subject to the aspect ratio and design of the tissue chamber. This can sometimes be limiting while recreating heterotypic cell models across the barrier due to the discrete nature of the pores. Furthermore, a common procedure to load cells into the microfluidic tissue chambers involves the centrifugation of cells into the chamber or the application of a vacuum. The diffusion barrier plays an important role in the successful loading of the chamber during the loading as it acts as a burst valve due to the air-water interface, thus providing a

[1]Department of Bioengineering, University of California, Berkeley, CA, USA. [2]Department of Materials Science and Engineering, University of California, Berkeley, CA, USA. [3]Diabetes Center, University of California, San Francisco, CA, USA. [4]Present address: Biomaterials Research Center, Korea Institute of Science and Technology, Seoul, South Korea. ✉e-mail: kehealy@berkeley.edu

resistance during loading without cells escaping into the media channel. Thus, the geometric features of such microchannel arrays must accommodate a compromise between providing a hydraulic resistance for a successful filling of the tissue chamber and a diffusion barrier that allows biomolecular diffusion and the ability to create heterotypic models across the barrier.

In this article, we demonstrate the development of a pillar array barrier in an MPS device that combines the advantages offered by porous membrane and microchannel barriers. We show that this pillar array diffusion barrier allows precise control over the pore location and the porosity that can be accurately tuned. The barrier does not require cumbersome handling and assembly. Furthermore, the polymeric pillar template allows control of the burst pressure and diffusion by simply changing the height of the array during fabrication. We hypothesized that control of burst pressure and diffusion by the pillar barrier allows robust tissue formation and provides an avenue to create physiologically dense tissue that models in vivo tissue physiology.

We implemented this barrier design in a cardiac MPS. We first present the concept of the barrier design, the fabrication procedure, and provide estimates of the burst pressure of different configurations of barriers. The ability to control biomolecular diffusion via alteration of the geometric parameters of the barrier is demonstrated via computational predictions made via the finite element method (FEM). Next, the ability of the barrier to accommodate the aggregation and formation of physiologically dense microtissue is demonstrated. The barrier allowed the exchange of nutrients

to the microtissues at physiologically relevant dynamic flow. Lastly, we show how the pillar array can be used to form a barrier model that incorporates endothelial cells to "vascularize" a cardiac tissue, as well as incorporate other tissue MPS models, such as pancreatic islets.

## Results and discussion
### Design and fabrication of the barrier

The MPS barrier consists of an array of pillars (Fig. 1A) separating two compartments, i.e., the cell chamber and the media channel. The porosity of the barrier can be altered by changing the distance between two pillars, defined in Fig. 1A as pore size. In our design, the porosity is defined by the ratio of the volume of voids to the total volume. For example, for a simple rectangular space of 125 x 708 μm (see Supplementary Fig. 1B), altering the pore size from 8 μm to 2 μm changes the porosity from 31% to 19% in a single configuration where the number of pillars is kept the same (i.e., 8) and their pillar diameters are altered to achieve the pore size. It is noted that porosity can be altered in multiple ways by varying the number of pillars and pore size, determined via the pillar diameters. On the other hand, changing the pillar height allows altering the porous volume that controls the hydraulic resistance offered by the barrier to a biomolecule and/or cell. For example, keeping the pore size constant at 8 μm, the porous volume can be linearly varied by increasing or decreasing the pillar height. Thus, the pillar array barrier allows flexibility in tuning parameters such as porosity and hydraulic resistance that serve as a tunable barrier in an MPS.

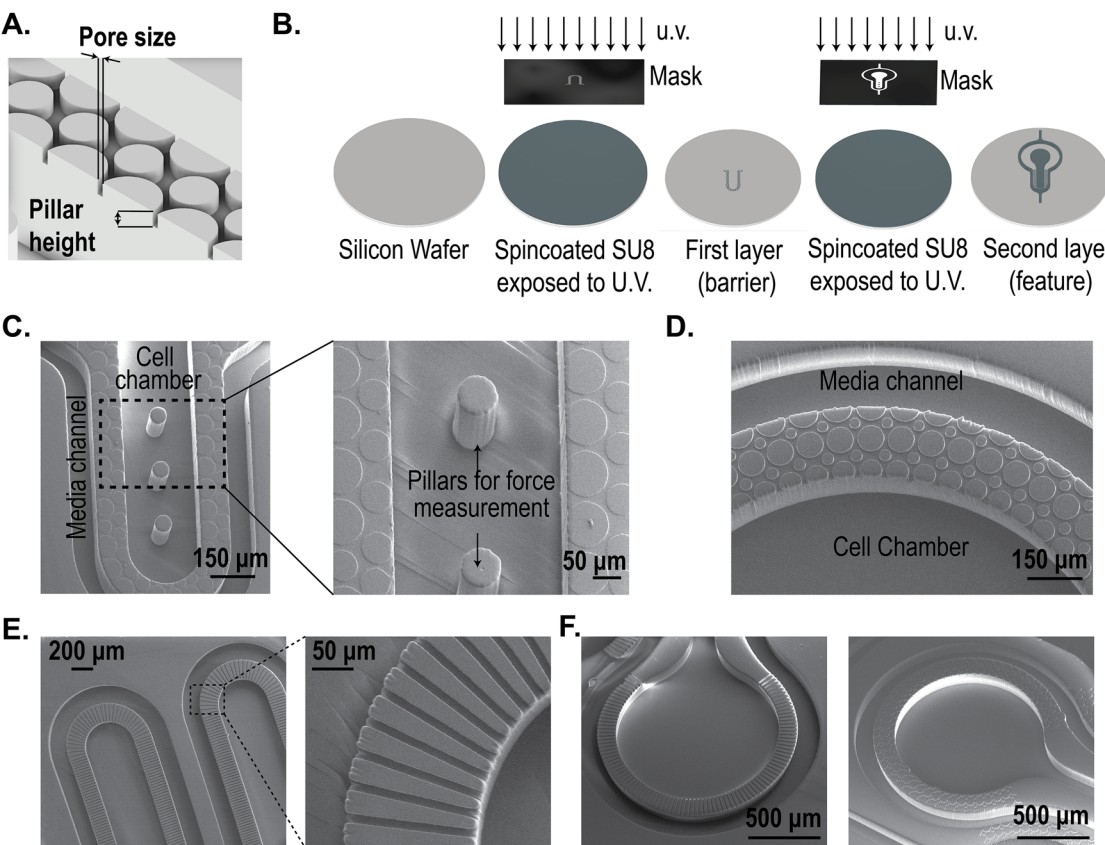

**Fig. 1 | Fabrication of pillar array barrier MPS device. A** An illustration of the replica molded pillar array barrier generated from the master mold. Pore size was defined in the mask design, and the thickness of the first layer of spin-coated photoresist determined pillar height. **B** Illustrated process to create a multi-layered master mold using SU8 photochemistry. The photoresist was spin-coated on a silicon wafer, and the first layer was generated by exposing it to UV light. After developing the first layer, fiduciary markers were used to create a second layer following a similar protocol of spin-coating and UV exposure. **C** SEM images of the PDMS replica molds of the cardiac MPS. A barrier separates the cell chamber and the media channel. Force-measuring pillars were fabricated within the cell chamber. Shown here is a barrier height of 2 μm, and cell chamber depth of 150 μm. **D** SEM image of replica mold with pillar array-based barriers of 10 μm pore size and 10 μm barrier height. The cell chamber and media channel are 100 μm in depth. **E** Example of a microchannel interface barrier in an MPS. **F** Comparison of a microchannel and a pillar array MPS interface barrier.

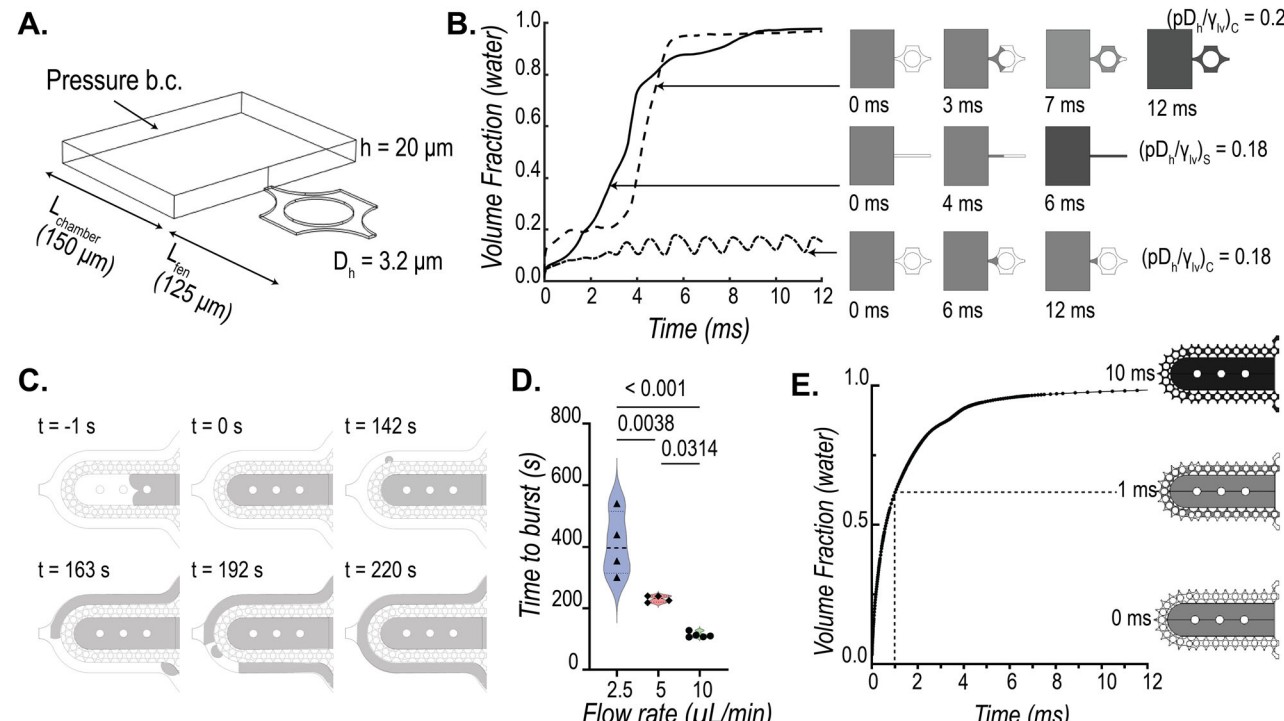

**Fig. 2 | Characterization of the burst pressure offered by the pillar array barrier.**
**A** A single fenestra finite element model was used to simulate the movement of the air-water interface across the microfluidic barrier to quantify burst pressure.
**B** Numerically estimated transient water volume fraction in the single fenestra models upon application of pressure (gray: water; white: air). The comparison shows the pillar barrier has a higher burst pressure than a single straight microchannel barrier for the same hydraulic diameter ($D_h = 3.2$ µm). Interfacial properties $\gamma_{lv} = 0.072 N/m$ and contact angle $\theta_c = 66°$ were used for the calculations.
**C** Representative sequences of experimentally captured movement of air-water

interface (gray: FITC-doped water; white: air) as water is pumped into the cell chamber in the MPS device. The time to first burst (shown at $t = 142$ s) is used to quantify the barrier function **D** offered at three different flow rates. **E** Finite element model predicted that the bursting takes place within fractions of a second when the pressure within the cell chamber reaches a critical value. The trace of water volume fraction shown here was obtained when the applied pressure at the cell chamber inlet was 6000 Pa. A fenestration height of 4 µm and a cell chamber height of 150 µm were used in this simulation.

We demonstrate two designs using different pore sizes and pillar heights in an MPS device. As an example, Fig. 1C shows the SEM of a cardiac MPS that follows a design form factor previously reported by our group[3,12,23]. The major components of the device are a central cell chamber, two adjacent media channels, and the pillar array barrier. The cell chamber has a width of 300 µm, while the media channels are 100 µm. The height of both the cell chamber and media channel is approximately 150 µm. The cell chamber is connected to the media channels by the pillar array barrier of height 2 µm and pore size of 8 µm. Details of the fabrication protocol are provided in the "Methodology" section. Figure 1D shows another form factor where the pillar height and pore size were 10 µm, and separate a media channel (100 µm width) and a circular cell chamber of 1500 µm diameter. In both designs, the pillar array acts as a barrier to aggregate cells within the cell chamber while loading and as a diffusion barrier that can model the endothelial compartment separating the microtissue from the perfusion medium, protecting the tissue itself from shear forces but allowing nutrient exchange. As a comparison to these pillar array interfaces, Fig. 1E, F show microchannel-based fenestrations[12,15].

In the next subsections, we report the characterization of the pillar array barrier in the cardiac MPS (Fig. 1C) via burst-pressure and biomolecular diffusion measurements.

### Pressure barrier offered by the pillar array diffusion barrier

The loading of cells into the cardiac MPS cell chamber involves centrifugation of singularized cells or spheroids suspended in a cell culture medium. For a successful aggregation of the cells/spheroids into a microtissue, the barrier must act as a burst valve to allow the cells to be contained within the cell chamber. During the cell loading process, the cell medium

replaces the air within the cell chamber and faces a high-pressure barrier at the opening of the pillar array due to an abrupt change in the cross-sectional area, stopping the liquid advancement. Thus, the diffusion barrier in the MPS acts as a capillary burst valve regulating the fluid flow. Empirically, the pressure required for the liquid to burst across a capillary burst valve ($\Delta p_b$) depends on the liquid–air surface tension ($\gamma_{lv}$) and the hydraulic diameter of the channel ($D_h$), such that $\Delta p_b \propto (\gamma_{lv}/D_h)$[24–26]. The expression of burst pressure depends on the capillary valve geometry as well. This forms the basis of the pillar array barrier, whereby variations in the circular geometry within a single fenestra achieve a higher burst pressure (Fig. 2A) than a single microchannel-based straight fenestra (Supplementary Fig. 1C).

To model the difference in resistance offered by a straight microchannel and a pillar array barrier, we implemented a level set method available via the COMSOL FEM package to simulate the air and water multiphase transport and calculate the burst pressures across different barrier designs. Figure 2A shows the computational domain used to simulate the air-water interface movement as we apply a constant pressure boundary condition to force water from a chamber (length: 150 µm, width: 200 µm, height: 20 µm) across a 125 µm length single fenestra of height 2 µm. The hydraulic diameters for the straight microchannel and pillar array fenestra were kept at 3.2 µm. For numerical stability and convergence, the values for the interfacial properties were kept constant for both geometries at $\gamma_{lv} = 0.072$ N/m and contact angle $\theta_c = 66°$. Figure 2B shows the transient volume fraction occupied by water in the fenestra as predicted by the level-set method for different applied pressures, normalized by $\gamma_{lv}/D_h$, to obtain a non-dimensional pressure $p^*$. As seen from Fig. 2B, a $p^*$ of 0.18 is enough to burst water out of the straight microchannel-based fenestra $((pD_h/\gamma_{lv})_S)$ but not sufficient to burst the valve formed by the pillar array. The pressure

required to burst the pillar array was 1.5 times that required for the microchannel fenestra, as seen from the $p^*$ of 0.27. The reason for this increased burst pressure can be explained by the pillar array fenestra consisting of a converging-diverging cross-sectional area that provides additional resistance for the moving liquid interface.

To validate the burst valve functionality of the pillar array experimentally, we recorded the movement of air–water interface as FITC-doped water was pumped into an empty cell chamber of the MPS. A representative sequence of these recordings is shown in Fig. 2C. We pumped water at flow rates of 2.5 μL/min, 5 μL/min, and 10 μL/min. Time to burst was defined as the first signal of water in the media channel. Based on this criterion, we see a clear barrier function offered by the pillar arrays as shown by the differential time to bursts at the three flow rates (Fig. 2D). We also conducted burst pressure tests with FITC-doped cell culture medium and saw a similar trend albeit with slightly different times owing to different fluid properties (Supplementary Fig. 1D). It is noted that as opposed to computational study, our boundary condition here is a mass inflow condition rather than a pressure boundary condition. This explains the different timescales estimated for the burst. The gradual increase of liquid mass within the cell chamber with the flow rates allows the build-up of pressure, and once a critical pressure is reached, the liquid bursts across the barrier. This takes a few hundred seconds with a flow inlet condition. A fluctuation of the liquid advancing front in the capillary valve, as recorded during our experiments, is provided as reference in the supplementary materials (Supplementary Movie 1). We see that when a high pressure of 6000 Pa is applied in our 3D level set model, the water interface bursts across the barrier without any fluctuations (Fig. 2E). Overall, we can infer that pressure builds up with the gradual influx of liquid in the cell chamber. The moving liquid front within the barrier fluctuates until critical pressure is reached, at which point the liquid floods spontaneously.

## Characterization of molecular transport across the diffusion barrier

Since altering the pore size and pillar height of the barrier changes the porous volume, it allows us to tune the diffusion across the barrier. The narrow cross-section of the barrier creates a fluidic resistance between the media channels and the cell chamber area. To characterize the transport of biomolecules, such as albumin, from the media channel into the cell chamber, we first obtained the steady-state velocity profiles within the MPS by solving the Navier-Stokes equation in the COMSOL FEM solver. Figure 3A shows the 3D computational domain and a representative velocity profile within the device when media is infused at 4800 μL/h (80 μL/min). The choice of flowrate is based values used in functional assessment of tissue in our previous studies.[1] Also shown are the 2D surface plots of velocity (magnitude) for a configuration consisting of a pillar barrier of 2 μm fenestration height and 8 μm pore size, and a configuration with no pillar barrier (i.e., a 2 μm microchannel connecting the media channel and cell chamber; no pillar). Surface plots were taken at $z = 1$ μm (i.e., half height of the barrier). The magnitude of velocities within the cell chamber was orders of magnitude lower (μm/s) when compared to those within the media channel (~mm/s). It is noted that the velocity with just a microchannel of 2 μm height but no pillars was significantly higher (3.45 μm/s) vs. a comparable configuration with pillars with 8 μm pore size (0.476 μm/s). This is expected as the pillars provide additional impediment to the flow streams. This affects the contribution of advective and diffusive rates of biomolecule transport from the media channel into the cell chamber, as described by the Peclet number (Pe = advective/diffusive rates). We summarize the Peclet number for different configurations of the pillar barrier and flow rates in the media channel (Fig. 3B). Figure 3B also provides the volume-averaged velocities estimated via FEM. A Peclet number of 1 represents equal contribution of advection and diffusion, whereas less than 1 implies dominance of diffusion in the mass transport of the biomolecule. It is clear that the introduction of pillar increases the diffusion transport of albumin into the cell chamber for 4800 μL/h when compared to a similar configuration without pillars (No pillar: 10.2 vs. pillar: 1.4). Fluorescence recovery after

photobleaching (FRAP) measurements made in rabbit ears by Chary and Jain have showed that albumin and biomolecular transport in tissue occur by both advection and diffusion[27]. Values of velocity in these tissues ranged from 0 to 2 μm/s (mean: 0.57; stdev: 0.15 μm/s), and Pe number 0.39 ± 0.14, indicative of a highly diffusion dominant transport[27].

To characterize the permeability of albumin through the barrier, we simulated the transport of 67 kDa albumin when injected into the media channel at a concentration of $c_0$ (7.46 nM) and a flowrate of 20 μL/h. The temporal evolution of albumin was obtained by coupling the Navier-Stokes solver in COMSOL to solve for the flow field and the Transport of Diluted Species solver to determine the concentration profiles. Figure 3C shows the transient evolution of the space-averaged value of albumin concentration in the cell chamber as a function of fenestration height when pore size was kept constant. The equilibration time varied from 12–48 h, within the order of time measured in rainbow trout tissues[28]. Based on these albumin transients, permeabilities were calculated at $0.5c_0$ for different configurations of the pillar barrier and flow rates in the media channel (Fig. 3D). As expected, configurations with lower Pe numbers had lower permeability value, with values plateauing when advective rates are negligible. Within these, certain configurations with 20 μL/h and 2 μL/h flowrates had permeability comparable to that measured in human tumor xenografts in mice measured $(6.06 \pm 4.30 \times 10^{-7})$ cm/s)[29]. Based on these and data presented in Fig. 3C, we used configuration with 2 μm fenestration height and 8 μm pore with flowrates of 20 μL/h and 2 μL/h. In these configurations, our pillar array barrier protects the tissue from the shear forces of the perfusion medium and allows nutrient exchange via a diffusion-dominant transport, modeling an endothelial barrier in vivo. The tunability of molecular transport offered by our pillar interface allows for modeling different disease states or accommodating the differential diffusion dynamics in different tissues[30–33].

## Leveraging a tunable diffusion barrier to design physiologically relevant cardiac microtissues

We used the pressure barrier functionality of the pillar array to aggregate hiPSC-derived cardiomyocytes in a cardiac MPS. The cell chamber has a width of 300 μm, while the media channels are 100 μm. The cell chamber and media channel heights are approximately 150 μm. The cell chamber consisted of 3 pillars (each 75 μm in diameter), separated by 225 μm (Fig. 4A). Singularized hiPSC-derived cardiomyocytes were mixed with cardiac-specific hiPSC-derived fibroblasts in a ratio of 80:20 to form a cardiac microtissue. Details of the differentiation and loading process are provided in Methodology. Briefly, cell mixtures of cardiomyocytes and fibroblasts were loaded into the cell chamber via centrifugation at 300 r.c.f. for 3 min. The aggregation of cells offered by the combination of cell chamber dimension, the pillars in the cell chamber, and the barrier function afforded via the pillar array leads to densely condensed cardiac microtissue in the MPS (Fig. 4A). The microtissue is 3D as validated by nuclei staining and high-resolution confocal imaging (Fig. 4A). We further quantified the number of cells and the volume of microtissue over an extended time (~months) and multiple biological replicates (Fig. 4B). The mean values for the number of cells and tissue volume were 3685 (coefficient of variation/cv: 0.40) and $1.91 \times 10^7$ μm³ (cv: 0.33), respectively. The cell density was calculated for each microtissue using the cell number and volume and had an average value of $1.95 \times 10^{14}$ cell/m³ (cv: 0.26; Fig. 4B), which matches the values observed in human heart muscle[34].

After condensation, the microtissues in our MPS had mean dimensions of 243.8 x 621.6 x 124.7 μm (cv: 0.05, 0.28, and 0.95, respectively). We wanted to predict whether these dimensions of the microtissue would lead to the formation of necrotic cores, given the high cell density, and whether the microtissue remains metabolically active and viable. Thus, we combined oxygen consumption measurements obtained from 2D culture with a reaction-diffusion finite element computational model.

Briefly, we conducted oxygen consumption measurements across different batches of differentiation using an Agilent Seahorse XFe96 machine. Traces of oxygen consumption rate (OCR) were obtained at the basal state and upon exposure to mitochondrial ATP synthase inhibitor

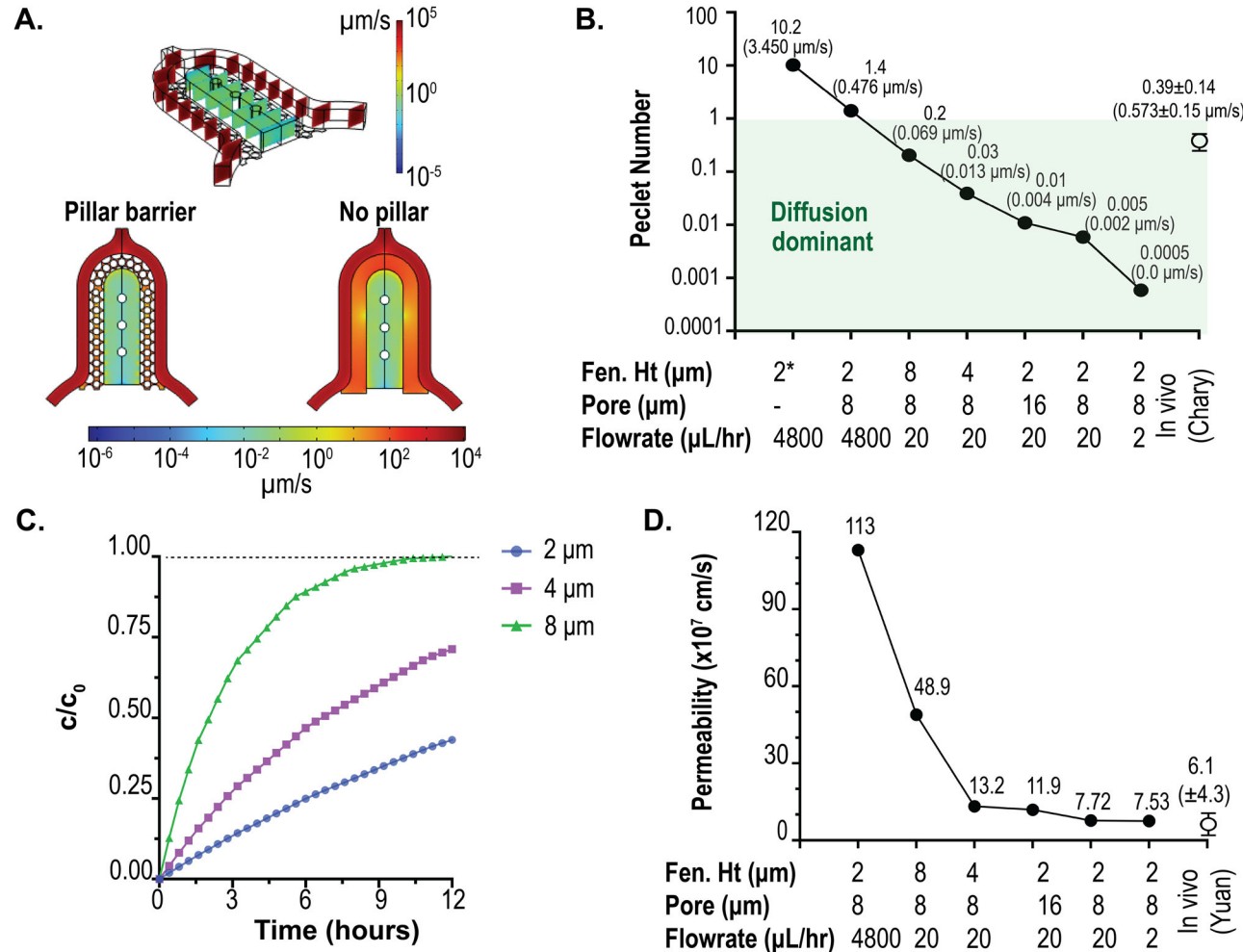

**Fig. 3 | Characterization of biomolecular diffusion across the pillar array barrier.**
**A** Steady state finite element model (FEM) prediction of velocity fields inside the cell chamber upon infusion of media via the media channel at 4800 µL/h. Shown is the comparison of velocity fields for a configuration with pillar barriers (2 µm fenestration height, and 8 µm pore size) and a no-pillar configuration with 2 µm channels connecting the media channel and the cell chamber. **B** FEM-predicted Peclet number for albumin diffusion for various combinations of fenestration height, pore sizes, and flow rates. Volume-averaged velocity for each configuration is provided in brackets. Peclet number for a no-pillar barrier configuration consisting of a 2 µm channel (no pores) is provided for reference, as well as Peclet number and velocity (mean ± standard deviation) based on in vivo data from Chary and Jain (1989)[27]. **C** Traces of albumin concentration in the cell chamber as a function of fenestration height obtained from FEM calculations. Flow rate within the media channel was set at 20 µL/h. **D** FEM-predicted permeability of albumin into the cell chamber via the pillar barrier as a function of fenestration height, pore size, and flowrates. Albumin vascular permeability (mean ± standard deviation) measured by Yuan et al. (1992)[29] demonstrates how our pillar array can be tuned to match in vivo data. (Abbreviations used: Fen. Ht: Fenestration height).

oligomycin, protonophore FCCP, and rotenone and antimycin (Rot+AA) that inhibit complex I and III activities, respectively (Fig. 4C)[35]. We measured a mean value of single-cell OCR (sOCR) of $4.67 \times 10^{-5}$ pmoles/s upon normalization with the number of cells per well. At the same time, we see a huge variation in this sOCR (cv: 0.69). Our mean and variation of the sOCR matches with data reported elsewhere in the literature on hiPSC-derived cardiomyocytes[36]. To account for this wide variation of sOCR, we implemented a population of models (PoM) approach to the reaction-diffusion model (detailed in Methodology). In this PoM, the reaction-diffusion model was solved 10,000 times, each time with a different value of sOCR that represented experimentally observed values. The computational domain (Fig. 4D) representing the tissue was based on measurements of the tissue geometry.

We chose a base geometry (V1) whose dimensions were 243.8 x 621.6 x 62.35 µm, which are within the ranges of dimensions observed for our microtissue. Next, we doubled the height of V1 to obtain another geometry V2 (243.8 x 621.6 x 124.7 µm). Since a higher variation was observed in the length of the tissue, we used another geometry (V3) of (288 x 950 x 124.7 µm). Overall, these geometries encompass the range of tissue volumes observed experimentally (V1: $0.8 \times 10^7$ µm³; V2: $1.6 \times 10^7$ µm³; V3: $3.1 \times 10^7$ µm³). The boundary condition (B.C.) imposed on the top surface of the tissue was a flux term, capturing the diffusion of oxygen via the PDMS[37]. A no-flux B.C. was imposed on the lower surface of the tissue facing the glass. The side walls of the tissue were imposed with a constant oxygen concentration B.C. of 0.21 mol/m³. The distribution of sOCR, used in the reaction term R (see "Methodology"), to simulate the statistical nature of the tissue OCR, is shown in Fig. 4D. The PoM produced profiles of oxygen within the tissue and the tissue OCR (or B distributions), defined as the surface integral of the oxygen flux into the tissue. Figure 4E shows the oxygen concentration distribution in tissue geometry V3 with sOCR prescribed at $6.1 \times 10^{-17}$ mol/s. Even with the higher end of the sOCR used, our PoM reveals that within the parameters used, the oxygen concentration profiles remain well above the critical value of 0.04 mol/m³ at which cells are believed to undergo cell death[38]. Supplementary

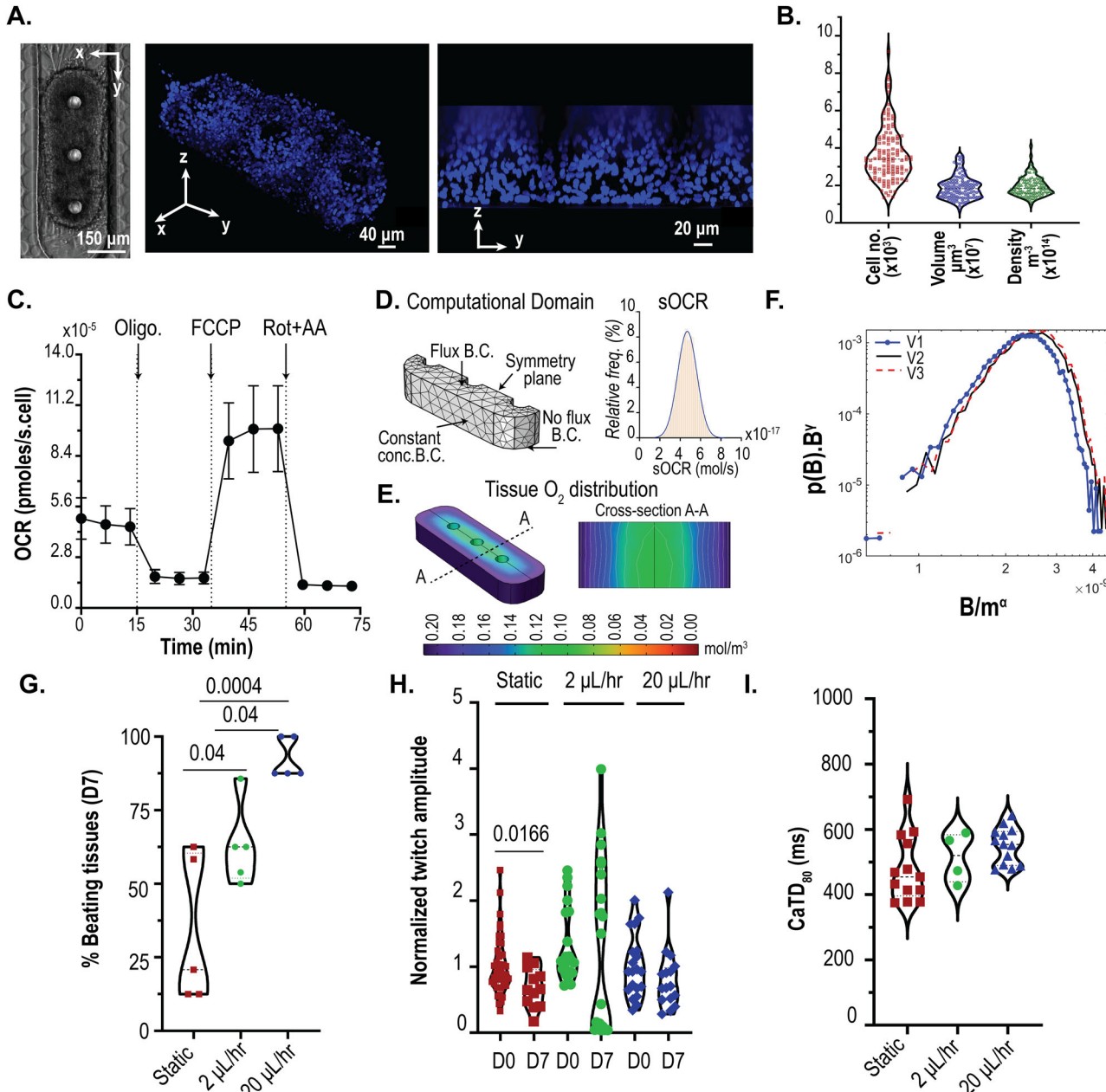

**Fig. 4 | Pillar array allows the generation of physiologically relevant microtissue.**
**A** 3D cardiac tissue formed within the MPS after aggregation of the single cells. Shown are the brightfield image and confocal images of the microtissue stained with nuclear stain DRAQ5. **B** Quantification of the cell number, volume, and density within the MPS. **C** Trace of mitochondrial oxygen consumption rates of cardiomyocytes in 2D obtained via Seahorse respirometer (n = 10; error bars represent standard error of mean). **D** Computational domain used for the FEM PoM to predict oxygen profiles within the 3D tissue, and distribution of sOCR utilized in the reaction term. **E** Representative image of 3D oxygen profile obtained using the PoM. Shown here is the geometry V2. **F** Log-log distribution of collapse of the tissue OCR to determine the relationship between tissue OCR (B distribution) and mass of microtissue m. $\gamma = 0.125$ and $\alpha = 0.8825\text{-}0.95$, as determined by finite size scaling. **G** Comparison of percentage tissues beating after day 7 in static vs. flow conditions. **H** Twitch amplitude comparison between day 0 vs. day 7. Significance between day 0 and day 7 was tested using Student's t-test. **I** Comparison of calcium transient duration $CaTD_{80}$ of microtissues on day 7 for static vs. flow conditions. (Oligo.: oligomycin, FCCP: carbonyl cyanide-p-trifluoromethoxyphenylhydrazone, Rot +AA: rotenone + antimycin A, B. C.: boundary condition, sOCR: single-cell oxygen consumption rate, $CaTD_{80}$: calcium transient duration at 80% repolarization).

Fig. 2A shows the tissue OCR probability distribution (p(B)) for V1, V2, and V3. As expected, tissue OCR increased with an increasing volume of the tissue. To investigate how these probability distributions scale with size, we assumed the probability density function suggested by Zaoli et al[39]. and shown in Eq. 1 as follows:

$$p\left(B, |, \langle m \rangle, \beta\right) = B^{-\beta} F\left(\frac{B}{\langle m \rangle^{\delta}}\right) \tag{1}$$

where $\delta$ is the scaling exponent defining the relation between volume (or masses; m) and tissue OCR (B), and F is a general scaling function. The exponent $\beta$ is a normalization exponent and is typically set to one[40]. To see how these tissue OCRs scale with respect to mass, we performed a finite-size scaling collapse of the distributions[41,42]. To measure the statistical distance between probability distributions, we leveraged the concept of probability contiguity as suggested by Bhattacharjee and Seno[41,42], and determined the scaling factor $\delta$ via a differential evolution optimization[43] which leads to the best collapse of the OCR (B distributions) for V1, V2, and V3. The collapse

reveals a non-isometric scaling of tissue OCR ($\alpha = 0.8825$) with respect to mass. The non-isometric scaling is indicative of near-diffusion limited tissue that are supposed to closely mimic physiological metabolism[44–46].

Based on the OCR profiles predicted by our model, the microtissue should not undergo functional decay if there is sufficient diffusion of nutrients into the tissue. We further performed live-dead (calcein AM + Ethidium homodimer I) assay on our tissues at static culture (i.e. no flow in the media channel) on day 7, and did not see any necrotic core formation (Supplementary Fig. 2C). However, to test if there were any differences in tissue behavior due to varying diffusion dynamics of nutrients (per Fig. 3), we compared our microtissues cultured in static vs. under dynamic medium perfusion at 2 and 20 μL/h. The value of 20 μL/h was chosen so that the shear stress at the walls was ~1 dynes/cm$^2$, which is considered near physiological[47,48]. The hiPSC-derived cardiac tissue exhibits automaticity (i.e., spontaneous beating without electrical stimulation)[49]. To test the functionality of the microtissue, we used two metrics: twitch amplitude and calcium transient duration at 80% repolarization percentage (CaTD$_{80}$). Twitch amplitude is associated with the contractile nature of the cardiac tissue, i.e,. how much the tissue contracts during a beat. The CaTD$_{80}$ provides a proxy measurement of the membrane potential waveform of the cardiac tissue[49]. We used published methods to measure the twitch amplitude and CaTD$_{80}$ of the microtissues[3,23,50]. We monitored the beating of the hiPSC-derived cardiac tissue over 1 week/7 days. Microtissues cultured in static conditions were fed every 2 days with fresh medium. Across different batches of measurements, 2/3rd of the microtissues cultured in static conditions failed to beat, indicative of functional alteration (Fig. 4G). In contrast, 92% of the microtissues in dynamic perfusion at 20 μL/h had beating, indicative of the functional integrity over 7 days. The fraction of beating tissues cultured with 2 μL/h was approximately 67%. We hypothesize that the varying fraction of beating tissues across these conditions was primarily driven by diffusion dynamics and advective current of biomolecules across the barrier interface as predicted by Fig. 3B. These microtissues with beating at day 7 were considered functionally active tissues. Within these functionally active tissues, static culture led to the peak twitch amplitude of the microtissue being slightly reduced on day 7 (Fig. 4H), whereas there was no statistical difference between the microtissues on day 0 vs. day 7 in MPS, where there was dynamic perfusion. The microtissues cultured in flow had higher CaTD$_{80}$, albeit not statistically significant compared to static cultured microtissues, but had much less variance (Fig. 4I). Thus, the pillar array allows the diffusion of nutrients and metabolites to maintain the functionality of physiologically dense 3D cardiac tissue.

## Pillar array as an engineered interface for creating heterotypic multi-tissue models

The pillar array can be leveraged to create an interface between endothelial cells (ECs) and the perivascular/parenchymal cardiac tissue (Fig. 5A). Such heterotypic models usually require scaling of the different tissues and/or providing physiologically meaningful fluidic shear stress[51]. To create a design that allows the scaling of two compartments with the pillar array as an engineered interface, we incorporated a modification of the two-step photolithography process for a three-step photolithography fabrication process (Fig. 5B). Briefly, we spun coat a silicon master wafer with a desired thickness of photoresist. This thickness is the target pillar height for the barrier. After exposing the photoresist to patterns incorporating the pore size, the layer was developed to incorporate the barrier design on the master wafer. Next, another layer of photoresist was used to generate the second layer (i.e., media channel) microstructure via the use of an alignment marker. After developing this layer, the third layer (i.e., cell chamber) microstructure was generated. The SEM for the master mold using this recipe is shown in Fig. 5C. In this study, we created a barrier height of approximately 10 μm, which will allow cells to migrate between the media channel and the cell chamber. The media channels were approximately 50 μm, and the cell chamber was approximately 100 μm. The corresponding polymeric PDMS replica mold with a pillar height of approximately 10 μm obtained from the master mold is shown

in Fig. 5D. The multi-layered fabrication demonstrated here will allow us to scale tissues and cell chambers to create heterotypic cell models in the future. To demonstrate one such heterotypic model, we created cardiac tissue consisting of cardiomyocytes and fibroblasts with ECs surrounding the cell chamber.

We cultured human coronary artery endothelial cells (HCAECs) in the media channels of the cardiac MPS. Cells were loaded and cultured in static overnight, after which fluid flow was introduced in the media channel. The flow rates were slowly ramped such that the shear stress experienced by the HCAECs for the first 12 h was at ~0.5 dynes/cm$^2$, the next 24 h at ~1 dynes/cm$^2$, and finally at ~2 dynes/cm$^2$ for the remaining 36 h. Representative images of the HCAECs cultured under dynamic fluid flow conditions for 3 days in the cardiac MPS are shown in Fig. 5E, F. We performed confocal imaging to assess the HCAECs coverage within the channel (as envisioned in Fig. 5A) and whether the endothelial cells covered the pillar barrier openings in the z-direction to add an active element to the passive barrier. We observed that the HCAECs covered the fenestrations, but did not form a 3D lumen covering all 4 walls of the channel (Fig. 5G). The pillar barrier allows the endothelial cells to form a relatively contiguous interface (Fig. 5F) that covers the pillar barrier openings in the z-direction, allowing the formation of an active component of the barrier. We performed further functional assessment of the barrier provided by the endothelial cells by infusing Alexa Fluor 594-tagged bovine serum albumin (BSA-AF594) through the media channels and measuring the leakage into the cell chamber following a protocol published in the literature[52]. We saw significant transport of the albumin into the cell chamber in devices without endothelial cell coverage versus those with a "vascularization" of the media channel (Fig. 5H). We next demonstrated that the MPS platform, consisting of the pillar array barrier, could be leveraged to create heterotypic cardiac models. Briefly, we introduced cardiomyocytes and fibroblasts into the cell chamber and allowed aggregation for a day. The following day, we functionalized the media channels with fibronectin, followed by the introduction of HCAECs. Figure 5I shows the heterotypic cardiovascular model formed within the MPS after 3 days in culture. In vitro cardiac MPS, consisting of heterotypic cellular components such as endothelial cells and macrophages, has been proposed to create more physiologically relevant organ models and remains the scope of future investigation leveraging our MPS platform with a tunable barrier interface. Figure 5J shows the SEM of the master mold of a device with microwells in the cell chamber, and pillar array separating the cell chamber and media channels. We introduced singularized stem cell-derived islet β cells into the cell chamber and allowed in-situ clustering following our published protocol[1]. Within 3 days, we see robust clustering of islet cells into clusters expressing insulin, demonstrating the pillar array's versatility to achieve different structural requirements (such as clustering of cells to organoids/spheroids) and use it for different tissue types.

In this study, we have demonstrated a highly tunable pillar interfacial barrier for MPS that can be leveraged for multi-tissue/cellular models. We haven't addressed issues of scaling of tissues/cells with respect to each other, or the design of a common medium required for functional coupling of cells/tissues. These remain challenges within the literature, and scope for future study. Finally, while we have shown the ease of fabrication of these pillar interfacial barriers with PDMS and replica molding, other fabrication strategies, such as embossing/thermoforming with materials like cyclic olefin or polycarbonate, will need future investigations.

## Conclusion

We report pillar arrays as tunable MPS interfacial barriers that allow control of biomolecular diffusion and support physiologically relevant tissue. Our results highlight the potential of pillar array interfaces to create heterotypic and multi-tissue MPS, paving the way for sophisticated drug and therapy discovery models.

## Methodology

**Fabrication.** The cardiac MPS was fabricated using photolithography and casting of polydimethylsiloxane (PDMS). Briefly, the design of the

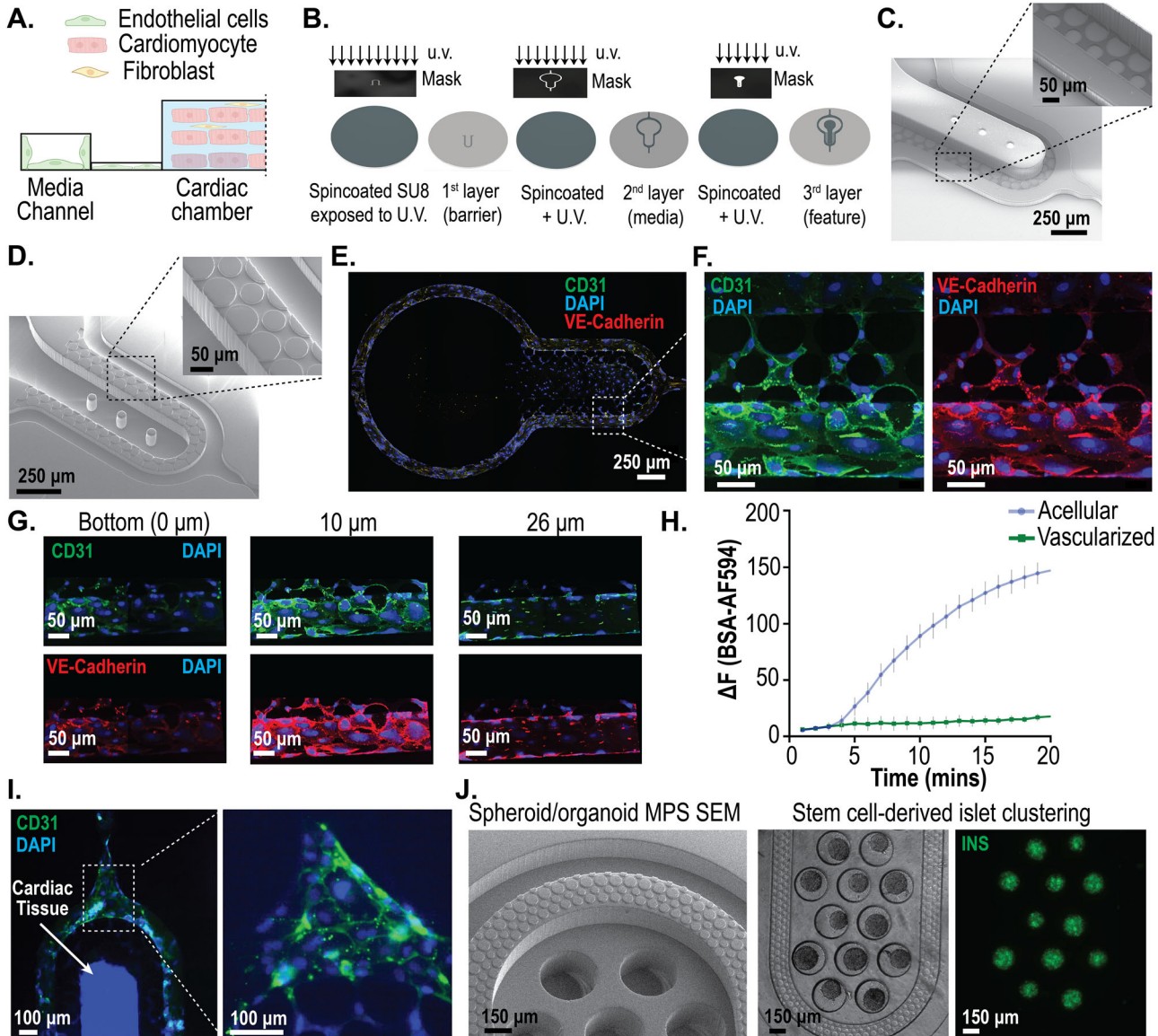

**Fig. 5 | Engineering heterotypic cell models leveraging the interface created by the pillar array barrier. A** A two-compartment model of endothelial cells and cardiac tissue, whereby the pillar array forms the engineered interface. **B** Fabrication schema for a two-compartment model with differential heights of the media channel and the cardiac chamber using a three-step/layer photolithography process to create master molds. **C** SEM image of the master mold with differential heights of SU8 microstructures for the two-tissue model. The media channel and cell chamber heights are 50 μm and 100 μm, respectively. The pillar height shown is 10 μm. **D** SEM image of PDMS replica mold obtained from the master mold. **E** Representative images showing endothelial cells forming a monolayer in the media channel cultured under near-physiological flow rates for 3 days. **F** High-resolution confocal image of endothelial cells from the inset in (**E**), showing the ability to form a cell layer across the pillar array barrier. **G** Confocal images of the vascular compartment at different heights show the coverage of the fenestration and media channel wall with endothelial cells. **H** BSA-AF594 fluorescence intensity in the cell chamber over time in the permeability assay for both acellular and vascularized MPS, demonstrating the barrier function offered by the endothelial cells. Traces represent mean, and error bars represent standard error of mean (*n* = 3). **I** Representative image of cardiac tissue, consisting of cardiomyocytes and fibroblasts, surrounded by endothelial cells in the media channel. **J** Adaptation of the pillar array to create stem cell-derived islet clusters in microwells in the cell chamber.

MPS device was made using a computer-aided design software package (AutoCAD, Autodesk Inc., San Rafael, CA). The design consisted of two layers: the fenestration layer and a feature layer consisting of the cell chamber with media channels. These designs were used to create two 5" x 5" laser-plotted photomasks, which were emulsions printed onto a transparency, with a resolution of 50,800 DPI (EMS Thin Metal Parts, Colorado Springs, CO). For photolithography, a 100 mm diameter Si wafer was first cleaned using piranha solution (1:3 v/v mix of $H_2SO_4$:$H_2O_2$), followed by spin-coating of a thin layer of negative photoresist (SU8 2002/2005, Kayaku Advanced Material, Westborough, MA) whose height was equal to that of the desired height of the fenestration layer. After soft baking, the photoresist was exposed to UV light using the fenestration photomask, followed by a post-exposure bake. The resist was then developed in propylene glycol methyl ether acetate, followed by a hard bake at 180 °C. A next round of spin-coating of a thicker negative photoresist (SU8 2100, Kayaku Advanced Material, Westborough, MA) was then performed for a desired height of the cell chamber. After soft baking, the Si wafer was aligned to the feature layer photomask using fiduciary markers to align the fenestration and cell chamber layers. The photoresist was exposed to UV light and then developed after a post-exposure bake. The process of hard baking was performed, followed by passivation of the Si mold by exposing it to

trichloro [1H, 1H, 2H, 2H-perfluorooctyl] silane (Sigma-Aldrich, catalog 448931) overnight via vapor deposition. This master mold was used to obtain replica molds of PDMS by pouring 20 g of a de-gassed 10:1 mixture of PDMS oligomer and cross-linking agent (Sylgard 184, Dow Corning, Midland, MI) and curing for 8 h at 65 °C in an oven. PDMS stamps were peeled off after cooling to room temperature, and 0.75 mm holes were punched at the loading and media port positions using a biopsy punch (Ted Pella, Redding, CA). PDMS stamps were then exposed to oxygen plasma (Plasma Equipment Technical Services, Livermore, CA) for 60 s (power: 21 W; flow: 98.8 sccm; pressure: 20 mTorr) and bonded to glass to create the device.

**SEM characterization.** Gold/palladium was deposited on the surface of PDMS samples for 100 s using the sputter coater (Cressington, UK). FEI Quanta 3D FEG SEM (Field Electron and Ion, Hillsboro, OR) was used to acquire images.

**Finite element model.** We used COMSOL 6.2 (COMSOL, Inc., Burlington, MA) to create finite element models (FEMs) of our cardiac MPS. We used FEM to quantify burst pressure in the diffusion barrier, the biomolecular transport, and the metabolism of the cardiac tissue within the MPS. Details of the FEM for each of the characterizations are provided in the following.

**Burst pressure calculations.** We implemented a level set method available via the COMSOL FEM package to simulate the air and water multiphase transport and calculate the burst pressures across different barrier designs. Geometries used for the simulation are shown in Fig. 2A, E. We implemented a pressure boundary condition at the inlet to simulate the required pressure field to burst the capillary valve formed by the diffusion barrier. Outlet gauge pressure was set to 0 Pa. The properties for air and water were imported from the built-in material library within COMSOL. Details of the implementation within the software can be found in the documentation from COMSOL Inc. A laminar flow solver for the Navier-Stokes equations incorporating surface tension forces was used to estimate the transport of mass and momentum for the two phases. Slip boundary condition was implemented by providing the contact angle between the material consisting of the walls (e.g., PDMS) and water using the Wetted Wall coupling feature in the software. Note that the contact angle of the solution domain boundaries can be determined experimentally by measuring the advancing contact angle in a goniometer. The multiphysics coupling between the level set and laminar flow solvers for the calculation of the water-air interface was performed via a segregated solver that involved a sequential solution of the two solvers.

**FEM characterization of biomolecular diffusion across barrier.** Transient flow profiles of candidate biomolecule (albumin; $D = 9.3 \times 10^{-7}\ cm^2/s$)[27] within the MPS were estimated through the solution of the incompressible Navier-Stokes equation utilizing the Laminar Flow module. Similarly, transient concentration profiles were estimated via the deployment of the Transport of Diluted Species module. The two module solvers were coupled via the Multiphysics solver Reacting flow, Diluted species. For the fluid flow solver, the boundary conditions were set as mass flow rate, and the outlet was set as a pressure outlet boundary, while a no-slip condition was set for the rest of the boundaries. For the calculation of concentration profiles, inlets were prescribed with a concentration of the candidate biomolecule, and outlets were set as outflow boundary conditions, while a no-flux condition was set for the rest of the boundaries.

**In Silico generation of cardiac tissue metabolic profiles in the MPS.** We implemented reaction-diffusion FEM to predict the oxygen profile within the cardiac microtissue. Specifically, the reaction-diffusion equation was of the form:

$$\frac{\partial c}{\partial t} = \nabla(D\nabla c) + R \tag{2}$$

In this Eq. 1, c is the concentration of oxygen, $D$ is the diffusion of oxygen in tissue, $\nabla$ the standard del operator ($\nabla \equiv i\frac{\partial}{\partial x} + j\frac{\partial}{\partial y} + k\frac{\partial}{\partial z}$), and R is the reaction rate that captures the oxygen consumption. In our study, we modeled R based on a Michaelis-Menten type schema, i.e.,

$$R = -\frac{sOCR \times \rho_c \times c}{k_m + c} \tag{3}$$

The values for single-cell oxygen consumption rate (sOCR) and cell density ($\rho_c$) were obtained from experimental measurements ("Results and Discussion"). The value of the Michaelis-Menten constant ($k_m$) was set at $6.9 \times 10^{-3}$ mol/m³, based on the value used for cardiac tissue elsewhere in the literature[53–56]. Upon solving for steady-state profiles of oxygen in the microtissue, the OCR of the tissue was calculated as the surface integral of the inward flux of oxygen at the boundaries. To implement the inherent variability in cell metabolism arising from intraspecific differences or batch-to-batch variation of stem cell differentiation to cardiomyocytes, we implemented a population of models (PoM) approach whereby we imposed variation in sOCR based on experimentally observed values. The PoM approach allowed capturing the distributions of tissue OCR (B distributions) as a consequence of the variability in single-cell metabolism (sOCR).

**Cardiac differentiation, culture, and cell loading into MPS.** Cardiomyocyte cells were derived from human induced pluripotent stem cells (hiPSC). The hiPSC line WTC-11 was expanded on growth factor-reduced Matrigel-coated plates (Corning, 354248) in mTeSR1 Plus medium (Stemcell Technologies, 100–0276) that was changed daily, passaged at 80% confluency using Accutase (Thermo Fisher Scientific, A1110501), and plated at a density of 12,000 cells/cm². Cells were fed culture medium supplemented with 5 µM Y-27632 dihydrochloride (Biogems, 1293823) for the first 24 h after passaging. Once confluent, the hiPSC cells were differentiated into human iCMs utilizing a chemically defined cardiomyocyte differentiation protocol with some modifications (https://doi.org/10.1073/pnas.1200250109). Briefly, hiPSCs were treated with 6 µM CHIR99021 (Biogems, 2520691) for 2 days in RPMI 1640 (Thermo Fisher Scientific, 11875119) with B27-insulin (Thermo Fisher Scientific, A1895601). The cells were subsequently treated with Wnt inhibitor IWP4 (Biogems, 6861787) in RPMI/B27 for another 2 days. Between 5–11 days of differentiation, RPMI/B27 medium was used and changed every other day. A robust spontaneous contractile activity was typically observed on days 8–10 of differentiation, at which the medium was switched to RPMI/B27+insulin (Thermo Fisher Scientific, 17504044). Cardiomyocyte purity was characterized using flow cytometry for cardiac troponin T (cTnT).

Once spontaneous contractile activity was observed, the hiPSC-CMs were dissociated with TrypLE 10x (Thermo Fisher Scientific, A1217703) and suspended in EB20 medium supplemented with 10 µM Y27632. To create isogenic iCM-icFb micromuscles in the MPS, we purified the iCMs by replating at a density of 100,000 cells/cm2 onto Matrigel, culturing in RPMI/B27+ without glucose (Thermo Fisher Scientific, 11879020) with 5 mM sodium Lactate (Sigma Aldrich, 71718) for 4 days. Cells were allowed to recover in RPMI/B27+ for 2 days. In parallel, we generated hiPSC-derived cardiac fibroblasts following a previously published protocol (https://doi.org/10.1007/7651_2020_300). On days 12–14, iCMs were dissociated using TrypLE10X, and icFbs were dissociated using Accutase. An isogenic cardiac microtissue was created by mixing 80% iCM-20% icFb EB20 by suspending cells with a density of approximately $2 \times 10^6$ cells/mL. 15,000 cells/8 µL that was injected into the loading port of each MPS. After 3 min of centrifugation at 300 g, MPS were inspected under the microscope. Chambers that were not filled with iCMs at this point were discarded. MPS were fed with 200 µL

EB20 medium supplemented with 10 μM Y27632 into the inlet tip, and gravity allowed for constant flow to the outlet until equilibrium was reached. The following day and every day from then on, the medium was changed to 80% RPMI/B27 +, 20% FGM3.

**Image acquisition for calcium transient and contractile activity studies.** Calcium transient and beating physiology studies were conducted using high-resolution image acquisition techniques. Cardiac MPS (microphysiological systems) were maintained at 37 °C during imaging, using a Tokai Hit stage with integrated heating. Spontaneous recordings included 6-s fluorescent videos (using the GCaMP6f WTC hiPSC line) for calcium transient analysis and 6-s brightfield videos to evaluate contractile activity. Post-experiment analysis was performed using a custom Python library, developed in-house, capable of processing fluorescence intensity over time and quantifying contractile motion from the brightfield recordings. Imaging was carried out with a NIKON TE300HEM microscope paired with a HAMAMATSU C11440/ORCA-Flash 4.0 digital CMOS camera, capturing videos at 100 frames per second (FPS). For fluorescence imaging, the Lumencor SpectraX Light Engine (Beaverton, OR) was employed in combination with a QUAD filter from Semrock (IDEX, Rochester, NY). Video acquisition was managed using Nikon's NIS-Elements software.

**Endothelial cells culture and cell loading into MPS.** Human coronary artery endothelial cells (PromoCell, Heidelberg, Germany) were cultured in the endothelial cell growth medium MV containing the supplement kit (PromoCell, Heidelberg, Germany), maintained at 37 °C in a 5% CO2 incubator. The MPS was functionalized with 0.5 mg/mL of fibronectin (Sigma, Burlington, MA) for 1 h at 37 °C before loading cells. A concentration of $30 \times 10^6$ cells/mL was loaded into the media ports of MPS using a pipette. After 1–2 h, non-attached cells were washed out by flushing medium. For culturing within the MPS, the medium used was 80% RPMI/B27 +, 20% FGM3 supplemented with 1 ng/mL VEGF.

**Immunofluorescent imaging.** MPS were flushed with PBS via the media channel for 10min, after which tissues were fixed with 4% paraformaldehyde for 15 min exposure, followed by PBS wash (2x). For staining the cardiac tissue, the devices were cut clean using a scalpel to expose the tissue, which is still attached to the PDMS. Following this tissues were stained by submerging the PDMS and tissue in different staining solutions. Tissues were first blocked with blocking buffer (1% BSA, 10% FBS, 0.5% Triton, 0.05% sodium azide) overnight at 4 °C. The next day, they were submerged in primary antibodies (mouse anti α-actinin, Life Technologies 41811; rabbit anti-myosin light chain 2 V (MLC-2V), Proteintech 10906-1-AP) at a 1:100 concentration in blocking buffer for 48 h at 4 °C. Tissues were then washed twice at 25 °C in blocking buffer for 2–3 h and washed a third time at 4 °C overnight. The secondary antibodies (goat anti-mouse IgG Alexa 568 H + L, Life Technology a11004; goat anti-rabbit IgG Alexa 488 H + L, Life Technology a11008) along with 1:600 DRAQ5 (Abcam, ab108410) were incubated in blocking buffer for 24 h. Tissues were then washed twice at 25 °C in blocking buffer for 2–3 h and a third time at 4 °C overnight before tissues were imaged.

**Mitochondrial respiration measurements.** Oxygen consumption of hiPSC-derived cardiomyocytes were performed in a Seahorse XFe96 machine (Agilent Technologies, Inc., Santa Clara, CA) using previously reported protocol[35]. Briefly, approximately 20,000 cells were seeded in each well and incubated overnight in a cell culture incubator at 37 °C and 5% CO2. After overnight incubation, samples were incubated with the assay buffer (Seahorse XF base medium, 1 mM pyruvate, 10 mM glucose, and 2 mM glutamax) in a non-CO2 incubator for 45 min. After which, they were washed again, and mitochondrial respiration via oxygen consumption rate (OCR) was measured in the XFe96 plate reader. Measurements were made at the basal state for 15 min. Injections of 1.5 μM ATPase inhibitor oligomycin, 2 μM protonophore FCCP, and 0.5 μM mixture of ETC complex III inhibitor antimycin-A and ETC complex I inhibitor Rotenone were made to measure the mitochondrial response. All concentrations reported are the final concentrations in each well.

**Functional assessment of endothelial permeability.** The protocol was adopted from published literature[52]. Cell culture medium containing 50 μg/mL of BSA-AF594 (Invitrogen) was perfused for 20 min at 20 μL/h, and fluorescence within the cell chamber was captured.

**Statistical analyses and reproducibility.** The software GraphPad Prism (GraphPad Software, San Diego, USA) was used for statistical analyses. The statistical differences between multiple groups were compared using one-way analysis of variance (ANOVA) followed by post hoc Tukey HSD to find means that were significantly different from each other. Differences between the means of two sample data were tested by the Student $t$-test. Information about sample sizes and replicates is provided for each dataset presented.

### Reporting summary
Further information on research design is available in the Nature Portfolio Reporting Summary linked to this article.

### Data availability
The data of this study are available from the corresponding author upon reasonable request.

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

## Acknowledgements
This research was supported by NIH UH3DK120004 and the NSF Engineering Research Center for Advanced Technologies for Preservation of Biological Systems (ATP-Bio) NSF EEC 1941543.

## Author contributions
I.G. and K.E.H. conceptualized the study. I.G. conducted computational modeling and analysis. I.G. and Y.K. conducted microfabrication and characterization. G.N., B.S. and T.N. optimized the cardiac tissue composition within the MPS and the imaging. J.V. conducted the Seahorse experiments. I.G., Y.K. and K.Y. optimized endothelial cell loading in the media channels. S.A. and I.G. optimized the MPS loading and culture of stem cell-derived islets. I.G., Y.K., G.N., B.S., J.V., K.Y., T.N. and K.E.H. analyzed the data. I.G. drafted the manuscript, and all authors edited it.

## Competing interests
K.E.H. and B.S. have a financial relationship with Organos Inc., and hence may benefit from the commercialization of the results of this research. The remaining authors declare no competing interests.
