## [Transparent Peer Review file · Communications Engineering]

Pillar arrays as tunable interfacial barriers for microphysiological systems

Corresponding Author: Professor Kevin Healy

Version 0:

Reviewer comments:

Reviewer #1

(Remarks to the Author)

The manuscript presents a novel microphysiological system (MPS) incorporating a circular pillar array as a tunable interfacial barrier, addressing limitations of conventional diffusion barriers, such as porous membranes and microchannel arrays. The study demonstrates the design, fabrication, and application of this pillar array barrier in cardiac MPS, highlighting its tunability in terms of porosity, pore size, and hydraulic resistance. The authors validate the utility of this barrier through computational modeling and experimental analyses, demonstrating its ability to facilitate physiologically relevant cardiac microtissue formation and heterotypic vascularized models. The manuscript emphasizes its potential in disease modeling, drug screening, and tissue engineering.

While the manuscript introduces a promising innovation in MPS design, several aspects require clarification and enhancement to strengthen the work's impact and reproducibility. The following concerns pertain to the experimental validation, clarity of methodology, and the broader implications of the findings.

1. The manuscript emphasizes the advantages of the pillar array in facilitating cell aggregation and nutrient diffusion, yet it provides limited experimental data to validate how these attributes translate into enhanced cell viability or functional outcomes. For example, the authors claim that the barrier promotes the formation of microtissues but do not present direct comparisons to traditional systems in terms of tissue organization, functionality, or survival over time. To better address this limitation, the authors should include comparative quantitative data, such as viability assays, functional readouts (e.g., contractile force for cardiac tissues), or long-term stability metrics under identical experimental conditions using conventional diffusion barriers and the proposed pillar array.
2. Although the authors focus on cardiac tissue and vascularized models, the broader utility of the pillar array for modeling other tissue types or systems is underexplored. The manuscript provides little discussion on whether the same design principles are adaptable to tissues with different metabolic or structural requirements, such as hepatic, neural, or intestinal tissues. The researchers could enhance the manuscript by including simulations or pilot experiments demonstrating the pillar array's versatility across other tissue types. For example, they might show how adjusting pore size or flow rates could support hepatic spheroid cultures or neuronal organoids. Such insights would significantly broaden the system's appeal and applicability.
3. The computational modeling used to predict diffusion dynamics and flow behavior lacks a clear explanation of how parameter ranges were selected. Parameters such as diffusion coefficients, flow rates, and porosity values are critical to validating the physiological relevance of the system, yet their alignment with *in vivo* conditions is not thoroughly justified. To address this, the authors should provide a more detailed rationale for their parameter choices, referencing relevant literature or experimental data where applicable. Additionally, correlating the model predictions with experimental observations (e.g., nutrient diffusion or metabolite exchange) would improve confidence in the system's design and functionality.
4. While the manuscript briefly discusses endothelial cell coverage on the pillar array, it does not provide sufficient functional validation of the vascularized tissue models. Critical aspects, such as barrier integrity, permeability, and shear stress effects, are not discussed in depth. These metrics are essential for determining the physiological relevance of vascularized models. To strengthen this section, the authors should perform and report functional assays that evaluate the vascular barrier's performance. For example, measuring permeability using fluorescent tracers, assessing endothelial response to shear stress could validate the system's ability to mimic *in vivo* conditions.
5. While the paper discusses prior limitations of porous membranes and microchannel barriers, it lacks comprehensive citations comparing performance metrics with those of the proposed pillar array barrier. The discussion of prior limitations in

porous membranes and microchannel arrays is not fully supported by citations. The manuscript would benefit from a more comprehensive review of previous systems, including quantitative comparisons of their performance metrics.

6. The manuscript could benefit from a discussion section that transparently addresses the limitations of the system, such as scalability for multi-tissue models or challenges in parameter tuning. The manuscript does not explicitly address the limitations of the pillar array system, such as challenges in scalability or parameter tuning for different applications. A transparent discussion of these challenges would demonstrate the authors' awareness of their system's constraints and encourage future work.

Minor comments

1. The manuscript uses terms like "vascularization" and "heterotypic models" interchangeably, which could confuse readers. The authors should define key terms early in the manuscript or ensure consistent usage throughout. This would prevent ambiguity and improve the overall readability of the paper.

2. Some figures, Fig. 5G, lack scale bars or resolution clarity. Fig. 5F does not include labels for ICC.

Reviewer #2

(Remarks to the Author)

This manuscript describes an engineering approach that uses a micropillar array to create a diffusion barrier within a microfluidic device for effectively compartmentalizing the cell culture chamber and media perfusion channels. The authors successfully demonstrate the barrier's function and its capability for molecular diffusion, which are critical for cell loading and culture, respectively. These demonstrations are supported by fluid-based and cell-culture experiments and in silico analysis. Additionally, the manuscript presents experimental proof of concept demonstration by recapitulating the cellular heterogeneity found at the perivascular interface of cardiac tissue. Although the proposed approach is robustly demonstrated, the manuscript's impact could be enhanced by clarifying some discussions in the text and by refining the relevance of certain experiments.

First, I would like to provide some major comments on the manuscript.

1) Page 1, Line 33: In the introduction, the authors discuss technical challenges regarding the use of porous membranes. However, I would like to question the relevance of this discussion for several reasons. Both the proposed method and the porous membrane approach are used for compartmentalization within microfluidic devices, but they are designed for fundamentally different configurations. Specifically, the proposed method facilitates lateral compartmentalization, whereas porous membranes are typically employed for vertical compartmentalization, which is advantageous for studies involving biological processes like transcytosis. Moreover, without direct comparisons between these two methods within this manuscript, readers might still prefer porous membranes for emulating the perivascular region in various tissues, as they can create a closed endothelial lumen that accurately mimics vascular barrier functions. The manuscript critiques the limitations of porous membranes, such as variations in pore size and position. However, these features do not look like their limitations and many commercialized membranes offer uniform pore sizes and well-controlled positions, mitigating these concerns. A more thorough discussion or comparative analysis emphasizing why the proposed approach may be superior to the use of porous membranes, similar to the analysis provided for the microchannel array method, is needed to strengthen the manuscript's claims.

2) Page 5, Line 8: To analyze the burst valve function, the authors provided both in silico analysis and experimental analysis using FITC-doped water. Given that this is one of the important features of the proposed approach, which can enable to isolate cells within the cell culture chamber during cell loading, it might be better to use more predictive solutions like cell culture media instead of water. Cell culture media may show different fluid properties, potentially affecting the burst valve function.

3) Page 8, Line 17: To investigate whether the proposed approach led to the formation of necrotic cores, the authors carried out OCR analysis, which may enable the estimation of reduced metabolism and thereby indirectly assess the presence of necrotic cores. However, there are several direct methods to measure the formation of necrotic cores within engineered tissues, such as live/dead cell analysis. Including results from a live/dead cell analysis would provide a more precise and straightforward assessment of necrotic core formation resulting from the proposed approach.

4) Page 9, Line 17: The experiments and comparisons discussed in this section are unclear. This part is crucial as it demonstrates the diffusion capability of the proposed approach for maintaining the physiological functions of engineered microtissues, which is the primary aim of the proposed approach. While measuring twitch amplitude, CaTD80, and beating frequency are robust methods for assessing the function of cardiac microtissues, the comparison between a single perfusion rate and a static culture condition raises concerns regarding the relevance of this experiment to the primary aim. What I understand is that the focus of this experiment is not to examine the effects of shear stress on tissue functions. Additionally, the comparison under static conditions suggests inadequate diffusion of cell culture molecules in the proposed device. Including an additional experimental condition using another diffusion barrier method, such as the microchannel-based approach, would provide a meaningful comparison and help establish the advantages of the proposed method in maintaining in vitro microtissues.

Alternatively, if the authors present comparisons between varying flow rate conditions, it might be more meaningful as it would allow for an estimation of how the tunability of diffusion, a key advantage of this proposed approach, affects the culture of cardiac microtissues. However, the current comparison seems to primarily investigate how shear stress impacts the culture of microtissues.

5) page 11, line 34. 'We note that this is similar to any membrane-based barrier, where endothelial cells only cover the porous barrier [8, 46].' I am not sure whether this discussion is correct. What I have known about the two citations, the four surfaces of their vascular channels separated by porous membranes from other channels, are fully lined with endothelial cells. (in reference 8, figure 1C and in reference 46, figure 1a). Please look into these and provide correct discussions.

6) Page 11, Line 35: I am wondering if this discussion is also correct. The cellular merging observed in the area of the pillar array seems to result from the migration of endothelial cells toward the center culture chamber, rather than from intercellular interactions between endothelial cells from the media channel and cardiac myocytes from the center chamber.

7) Page 11, Line 38: The migration of endothelial cells into the pillar array area may limit the diffusion of culture media molecules between the center and media chambers. Thus, to demonstrate the co-culture capability of the proposed approach, specifically for co-culturing endothelial cells in the media channel with cardiac myocytes and fibroblasts in the center chamber, it is necessary to investigate the viability, twitch amplitude, CaTD80, and beating frequency of the cardiac microtissues over 7 days, similar to what was done in the previous section.

Minor comments.

1) Page 2, Line 13: 'which results in filling of the tissue chambers' is confusing. It seems it needs to be corrected to 'which results in filling of the media channels.'

2) In Figure 5F: Do red and green indicate VE-cadherin and CD31, respectively? Although Figure 5E provides indicators, it would be clearer if a dotted line extended from Figure 5E to Figure 5F, similar to what was done in Figure 5H.

3) In Figure 5G: It is unclear where the media channel and pillar array area are located in this figure. Why are endothelial cells shown on the 10 μm surface and not on the 0 μm surface? This figure seems to label the 26 μm for the top-surface of the media (vascular) channel, but why are there no endothelial cells there? This observation is related to the comment #5.

Reviewer #3

(Remarks to the Author)

The idea of generating microfluidic devices with tunable porosity is potentially interesting. However I see a major drawback related to the pore size of 8 μm , which looks to high for any type of biological barrier (standard membranes have pores around 0.45 μm). Indeed, as shown also by author the cells go through the barrier which is actually not really physiological. It would also be difficult to monitor barrier integrity with standard method such as TEER as well to do any significant evaluation regarding the passage of substances.

The author should justify better these issues.

Version 1:

Reviewer comments:

Reviewer #1

(Remarks to the Author)

Reviewer #2

(Remarks to the Author)

The authors addressed all of the concerns I raised by revising the manuscript, conducting additional experiments, or acknowledging them as limitations of the proposed approach.

Reviewer #3

(Remarks to the Author)

after revision the authors significantly improved the impact of the manuscript. I just suggest to include a discussion on the use of different pore size which is crucial for this study

We would like to thank the reviewers for their time to thoroughly review our paper and for the many useful comments and suggestions made. In response, we have made several significant improvements to our manuscript. We have included a revised manuscript with track changes. In addition, our detailed responses to the reviewers' comments are given below. The comments are in italics, our responses are given in regular font, specific text changes are indicated in red, and original text is indicated in blue. The page numbers included for changes made to the manuscript refer to the track changes version. References are provided in the rebuttal document for ease of reading and reflect the same reference number in the manuscript.

Reviewer 1

Comment 1: *Although the authors focus on cardiac tissue and vascularized models, the broader utility of the pillar array for modeling other tissue types or systems is underexplored. The manuscript provides little discussion on whether the same design principles are adaptable to tissues with different metabolic or structural requirements, such as hepatic, neural, or intestinal tissues. The researchers could enhance the manuscript by including simulations or pilot experiments demonstrating the pillar array's versatility across other tissue types. For example, they might show how adjusting pore size or flow rates could support hepatic spheroid cultures or neuronal organoids. Such insights would significantly broaden the system's appeal and applicability.*

Response: We thank the reviewer for the suggestion. We have added data showing the feasibility of the interfacial barrier to aid the clustering of stem cell-derived islet cells within microwells (revised Figure 5), showing the pillar array's versatility to achieve different structural requirements (such as clustering of cells to organoids/spheroids) and use it for different tissue types.

Page 13; line 4-11:

Figure 5J shows the SEM of the master mold of a device with microwells in the cell chamber, and pillar array separating the cell chamber and media channels. We introduced singularized stem cell-derived islet β cells into the cell chamber and allowed in-situ clustering following our published protocol.¹ Within 3 days, we see robust clustering of islet cells into clusters expressing insulin, demonstrating the pillar array's versatility to achieve different structural requirements (such as clustering of cells to organoids/spheroids) and use it for different tissue types.

Page 15; line 8-11:

The pillar array can also be used to create tissue barrier models, such as vascularized cardiac tissue, and leveraged to create other tissue types with different structural requirements (e.g. clustering of spheroids/organoids).

Figure 5: Engineering heterotypic cell models leveraging the interface created by the pillar array barrier. (A) A two-compartment model of endothelial cells and cardiac tissue, whereby the pillar array forms the engineered interface. **(B)** Fabrication schema for a two-compartment model with differential heights of the media channel and the cardiac chamber using a three-step/layer photolithography process to create master molds. **(C)** SEM image of the master mold with differential heights of SU8 microstructures for the two-tissue model. The media channel and cell chamber heights are 50 µm and 100 µm, respectively. The pillar height shown is 10 µm. **(D)** SEM image of PDMS replica mold obtained from the master mold. **(E)** Representative images showing endothelial cells forming a monolayer in the media channel cultured under near-physiological flowrates for 3 days. **(F)** High-resolution confocal image of endothelial cells from inset in (E), showing the ability to form cell layer across the pillar array barrier. **(G)** Confocal images of the vascular compartment at different heights show the coverage of the fenestration and media channel wall with endothelial cells. **(H)** BSA-AF594 fluorescence intensity in the cell chamber over time in permeability assay for both acellular and vascularized MPS demonstrating the barrier function offered by the endothelial cells. Traces represent mean and error bars represent standard error of mean (n = 3). **(I)** Representative image of cardiac tissue, consisting of cardiomyocytes and fibroblasts, surrounded by

endothelial cells in the media channel. **(J)** Adaptation of the pillar array to create stem cell-derived islet clusters in microwells in the cell chamber. INS: Insulin.

Comment 2: *The computational modeling used to predict diffusion dynamics and flow behavior lacks a clear explanation of how parameter ranges were selected. Parameters such as diffusion coefficients, flow rates, and porosity values are critical to validating the physiological relevance of the system, yet their alignment with in vivo conditions is not thoroughly justified. To address this, the authors should provide a more detailed rationale for their parameter choices, referencing relevant literature or experimental data where applicable. Additionally, correlating the model predictions with experimental observations (e.g., nutrient diffusion or metabolite exchange) would improve confidence in the system's design and functionality.*

Response: The reviewer raises a good point, and we have revamped Figure 3 to provide the rationale of the choice of flowrates, fenestration height, and pore sizes. We also provide calculated values from in vivo experimental datasets found in the literature to show alignment of these choices with in vivo conditions. The following are the changes made to the text and figure.

Page 6 line 23 – Page 8 line 21:

Characterization of molecular transport across the diffusion barrier. Since altering the pore size and pillar height of the barrier changes the porous volume, it allows us to tune the diffusion across the barrier. The narrow cross-section of the barrier creates a fluidic resistance between the media channels and the cell chamber area. To characterize the transport of biomolecules, such as albumin, from the media channel into the cell chamber we first obtained the steady state velocity profiles within the MPS by solving the Navier-Stokes equation in COMSOL FEM solver. **Figure 3A** shows the 3D computational domain and a representative velocity profile within the device when media is infused at 4800 $\mu\text{L/hr}$ (80 $\mu\text{L/min}$). The choice of flowrate is based values used in functional assessment of tissue in our previous studies.¹ Also shown are the 2D surface plots of velocity (magnitude) for a configuration consisting of a pillar barrier of 2 μm fenestration height and 8 μm pore size, and a configuration with no pillar barrier (i.e. just a 2 μm microchannel connecting the media channel and cell chamber; no pillar). Surface plots were taken at $z=1\mu\text{m}$ (i.e. half height of the barrier). Magnitude of velocities within the cell chamber were orders lower ($\sim \mu\text{m/s}$) when compared to the those within the media channel ($\sim \text{mm/s}$). It is noted that velocity with just a microchannel of 2 μm height but no pillars was significantly higher (3.45 $\mu\text{m/s}$) vs. a comparable configuration with pillars with 8 μm pore size (0.476 $\mu\text{m/s}$). This is expected as the pillars provide additional impediment to the flow streams. This affects the contribution of advective and diffusive rates of biomolecule transport from the media channel into the cell chamber, as described by Peclet number ($Pe = \text{advective/diffusive rates}$). We summarize the Peclet number for different configurations of the pillar barrier and flow rates in the media channel (**Figure 3B**). Figure 3B also provides the volume-averaged velocities estimated via FEM. A Peclet number of 1 represents equal contribution of advection and diffusion, whereas less than 1 implies dominance of diffusion in the mass transport of the biomolecule. It is clear that the introduction of pillar increases the diffusion transport of albumin into the cell chamber for 4800 $\mu\text{L/hr}$ when compared to a similar configuration without pillars (No pillar: 10.2 vs. pillar: 1.4). Fluorescence Recovery After Photobleaching (FRAP) measurements made in rabbit ears by Chary and Jain have showed that albumin and biomolecular transport in tissue occur by both advection and diffusion. Values of velocity in these tissues ranged from 0 to 2 $\mu\text{m/s}$ (mean: 0.57; stdev: 0.15 $\mu\text{m/s}$), and Pe number 0.39 ± 0.14 , indicative of a highly diffusion dominant transport.²²

To characterize the permeability of albumin through the barrier, we simulated the transport of 67 kDa albumin when injected into the media channel at concentration of c_0 (7.46 nM) and a flowrate of 20 $\mu\text{L/hr}$. The temporal evolution of albumin was obtained by coupling the Navier-Stokes solver in COMSOL to solve for the flow field and the Transport of Diluted Species solver

to determine the concentration profiles. **Figure 3C** shows the transient evolution of space-averaged value of albumin concentration in the cell chamber as a function of fenestration height when pore size was kept constant. The equilibration time varied from 12-48 hours, within the order of time measured in rainbow trout tissues.²³ Based on these albumin transients, permeabilities were calculated at 0.5c₀ for different configurations of the pillar barrier and flow rates in the media channel (**Figure 3D**). As expected, configurations with lower Pe numbers had lower permeability value, with values plateauing when advective rates are negligible. Within these, certain configurations with 20 μL/hr and 2 μL/hr flowrates had permeability comparable to those measured in human tumor xenografts in mice measured ($6.06 \pm 4.30 \times 10^{-7}$ cm/s).²⁴ Based on this and data presented in Figure 3C, we used configuration with 2 μm fenestration height and 8 μm pore with flowrates of 20 μL/hr and 2 μL/hr. In these configurations, our pillar array barrier protects the tissue from the shear forces of the perfusion medium and allows nutrient exchange via a diffusion-dominant transport, modeling an endothelial barrier in vivo. The tunability of molecular transport offered by our pillar interface allows for modeling different disease states or accommodate the differential diffusion dynamics in different tissues.²⁵⁻²⁸

Figure 3: Characterization of biomolecular diffusion across the pillar array barrier. (A) Steady state FEM predictions of velocity fields inside the cell chamber upon infusion of media via the media channel at 4800 μL/hr. Shown is the comparison of velocity fields for a configuration with pillar barriers (2 μm fenestration height, and 8 μm pore size) and a no pillar configuration with 2 μm channels connecting the media channel and the cell chamber. **(B)** Peclet number for albumin diffusion for various combinations of fenestration height, pore sizes,

and flowrates. Volume averaged velocity for each configuration is provided in brackets. Peclet number for a no pillar barrier configuration consisting of a 2 μm (no pores) is provided for reference, as well as Peclet number based on in vivo data from Chary and Jain (1989). **(C)** Traces of albumin concentration in the cell chamber as a function of fenestration height obtained from FEM calculations. Flow rate within the media channel was set at 20 $\mu\text{L/hr}$. **(D)** FEM-predicted permeability of albumin into the cell chamber via the pillar barrier as a function of fenestration height, pore size, and flowrates. Albumin vascular permeability measured by Yuan et al. (1992) demonstrates how our pillar array can be tuned to match in vivo data.

Comment 3: *While the manuscript briefly discusses endothelial cell coverage on the pillar array, it does not provide sufficient functional validation of the vascularized tissue models. Critical aspects, such as barrier integrity, permeability, and shear stress effects, are not discussed in depth. These metrics are essential for determining the physiological relevance of vascularized models. To strengthen this section, the authors should perform and report functional assays that evaluate the vascular barrier's performance. For example, measuring permeability using fluorescent tracers, assessing endothelial response to shear stress could validate the system's ability to mimic in vivo conditions.*

Response: We thank the reviewer for this suggestion. We have performed permeability studies using fluorescent albumin (BSA-AF594) dye to provide semi-quantitative measurement of vascular barrier. The added dataset is in panel Figure 5H found in the response in Comment 1. Following is the added text.

Page 12; line 36-41:

We performed further functional assessment of the barrier provided by the endothelial cells by infusing Alexa Fluor 594-tagged bovine serum albumin (BSA-AF594) through the media channels, and measuring the leakage into the cell chamber following protocol published in the literature.⁴⁷ We saw significant transport of the albumin into the cell chamber in devices without endothelial cell coverage versus those with a “vascularization” of the media channel (**Figure 5H**).

Comment 4: *The manuscript emphasizes the advantages of the pillar array in facilitating cell aggregation and nutrient diffusion, yet it provides limited experimental data to validate how these attributes translate into enhanced cell viability or functional outcomes. For example, the authors claim that the barrier promotes the formation of microtissues but do not present direct comparisons to traditional systems in terms of tissue organization, functionality, or survival over time. To better address this limitation, the authors should include comparative quantitative data, such as viability assays, functional readouts (e.g., contractile force for cardiac tissues), or long-term stability metrics under identical experimental conditions using conventional diffusion barriers and the proposed pillar array.*

Response: We did not make any claims about enhanced viability due to the pillar barrier design but rather a more sophisticated tunability offered by it. This has been demonstrated via Figures 2 and 3. While we have added viability data and comparative study across different flowrates within the pillar configuration to show the differences arising from biomolecular diffusion (see revised Figures 4 and S2), we don't believe comparative studies with other traditional systems with new experiments will be within the scope of the study given the limit on figures and length of the manuscript.

Comment 5: *While the paper discusses prior limitations of porous membranes and microchannel barriers, it lacks comprehensive citations comparing performance metrics with those of the proposed pillar array barrier. The discussion of prior limitations in porous membranes and microchannel arrays is not fully supported by citations. The manuscript would benefit from a more*

comprehensive review of previous systems, including quantitative comparisons of their performance metrics.

Response: We thank the reviewer for this suggestion. We have edited our manuscript and further added citations to literature detailing the limitations of the porous membranes. We have also added SEM images of commercially available membranes as a reference.

Page 1 line 42 – Page 2 line 6:

The fabrication process is scalable and allows for wide adoption. However, there is huge variation in pore sizes and pore position in these membranes, due to stochastic distribution of pores that overlap forming unwanted larger openings, inter-pore spacing, and pore densities (**Figure S1A**).^{18,19} Alternatively, others have created membranes out of replica molded polydimethylsiloxane (PDMS), electropun membranes, and silicon nitride films.^{11,20–22} The process of incorporating these membranes into a MPS involves functionalizing membranes/adhesive application and cumbersome alignment with the microfluidic chambers during assembly.^{1,2} Furthermore, membrane integrity and porosity may be compromised during solvent cleaning and handling during assembly.¹⁸

Figure S1: (A) SEM images of commercially available PET membrane showing pores that overlap forming unwanted larger openings, inter-pore spacing, and pore densities. (B) Variation of pore size as defined by the distance between the pillars changes the porosity of the pillar-

based fenestration layer. Shown here are two interfaces created in a 125 μm x 708 μm rectangle with 8 μm and 2 μm pore sizes and porosity of 31% and 19%, respectively. **(C)** A single microchannel-based fenestra finite element model was used to simulate the movement of the air-water interface across the microfluidic barrier to quantify burst pressure. **(D)** The time to first burst, used to quantify the barrier function, with FITC-doped cell culture medium at three different flow rates.

Comment 6: *The manuscript could benefit from a discussion section that transparently addresses the limitations of the system, such as scalability for multi-tissue models or challenges in parameter tuning. The manuscript does not explicitly address the limitations of the pillar array system, such as challenges in scalability or parameter tuning for different applications. A transparent discussion of these challenges would demonstrate the authors' awareness of their system's constraints and encourage future work.*

Response: We value the reviewer's comment. To keep within the word limit of the journal, we have briefly added a few limitations in line with the suggested areas highlighted by the reviewer.

Page 13; line 12-19:

In this study we have demonstrated a highly tunable pillar interfacial barrier for MPS that can be leveraged for multi-tissue/cellular models. We haven't addressed issues of scaling of tissues/cells with respect to each other, or design of a common medium required for functional coupling of cells/tissues. These remain challenges within literature, and scope for future study. Finally, while we have shown the ease of fabrication of these pillar interfacial barriers with PDMS and replica molding, other fabrication strategies such as embossing/thermoforming with materials like cyclic olefin or polycarbonate will need future investigations.

Comment 7 (minor): *The manuscript uses terms like "vascularization" and "heterotypic models" interchangeably, which could confuse readers. The authors should define key terms early in the manuscript or ensure consistent usage throughout. This would prevent ambiguity and improve the overall readability of the paper.*

Response: We have added definition of the word heterotypic in the introduction of the manuscript to avoid any confusion.

Page 1; line 35-38:

The porous membrane allows the diffusion of nutrients and molecules as well as allowing heterotypic cell-cell communication (i.e. communication between different types of cells in a tissue) via the incorporation of two different cell-types on either side of the membrane.

Comment 8 (minor): *Some figures, Fig. 5G, lack scale bars or resolution clarity. Fig. 5F does not include labels for ICC.*

Response: Thank you for pointing out this. We have fixed it in the next version of our manuscript.

Comment 1: *Page 1, Line 33: In the introduction, the authors discuss technical challenges regarding the use of porous membranes. However, I would like to question the relevance of this discussion for several reasons. Both the proposed method and the porous membrane approach are used for compartmentalization within microfluidic devices, but they are designed for fundamentally different configurations. Specifically, the proposed method facilitates lateral compartmentalization, whereas porous membranes are typically employed for vertical compartmentalization, which is advantageous for studies involving biological processes like transcytosis. Moreover, without direct comparisons between these two methods within this manuscript, readers might still prefer porous membranes for emulating the perivascular region in various tissues, as they can create a closed endothelial lumen that accurately mimics vascular barrier functions. The manuscript critiques the limitations of porous membranes, such as variations in pore size and position. However, these features do not look like their limitations and many commercialized membranes offer uniform pore sizes and well-controlled positions, mitigating these concerns. A more thorough discussion or comparative analysis emphasizing why the proposed approach may be superior to the use of porous membranes, similar to the analysis provided for the microchannel array method, is needed to strengthen the manuscript's claims.*

Response: We thank the reviewer for the suggestion. However, we would like to challenge the reviewer's view on the uniformity of pore size and well-controlled positions of pores in commercially available membranes. We have extensive experience in using these membranes (Goswami et al. Lab Chip 2022, Lee-Montiel et al. Frontiers Pharmacol. 2021, Qi et al. Nat. Comm. 2024). These membranes have stochastic distribution of pores that overlap, forming unwanted openings, varying inter-pore spacing, and pore densities. Others in the literature have also noted the same (e.g. Casillo et al. ACS Biomater Sci Eng. 2017). Furthermore, the membrane integrity and porosity may be compromised during handling and assembly, as has been our experience, as well as others in the literature (e.g. Bolze et al. Chem Engg. Tech. 2021). We have taken SEM images of commercially available PET membranes used in our previous studies (Goswami et al. 2022, Lee-Montiel et al. 2021) to provide the reviewer and the readers with the stochasticity of the pores in these porous systems (added Figure S1A). While porous membranes remain a viable option for tissue models, we have highlighted a few limitations with added citations.

Page 1 line 42 – Page 2 line 6:

The fabrication process is scalable and allows for wide adoption. However, there is huge variation in pore sizes and pore position in these membranes, **due to stochastic distribution of pores that overlap forming unwanted larger openings, inter-pore spacing, and pore densities (Figure S1A).**^{18,19} Alternatively, **others have created membranes out of replica molded polydimethylsiloxane (PDMS), electropun membranes, and silicon nitride films.**^{11,20–22} The process of incorporating these membranes into a MPS involves functionalizing membranes/**adhesive application** and cumbersome alignment with the microfluidic chambers during assembly.^{1,2} Furthermore, **membrane integrity and porosity may be compromised during solvent cleaning and handling during assembly.**¹⁸

Figure S1: (A) SEM images of commercially available PET membrane showing pores that overlap forming unwanted larger openings, inter-pore spacing, and pore densities. (B) Variation of pore size as defined by the distance between the pillars changes the porosity of the pillar-based fenestration layer. Shown here are two interfaces created in a 125 μm x 708 μm rectangle with 8 μm and 2 μm pore sizes and porosity of 31% and 19%, respectively. (C) A single microchannel-based fenestra finite element model was used to simulate the movement of the air-water interface across the microfluidic barrier to quantify burst pressure. (D) The time to first burst, used to quantify the barrier function, with FITC-doped cell culture medium at three different flow rates.

Comment 2: Page 5, Line 8: To analyze the burst valve function, the authors provided both *in silico* analysis and experimental analysis using FITC-doped water. Given that this is one of the important features of the proposed approach, which can enable to isolate cells within the cell culture chamber during cell loading, it might be better to use more predictive solutions like cell culture media instead of water. Cell culture media may show different fluid properties, potentially affecting the burst valve function.

Response: The point made by the reviewer is well taken, and we have redone the experiment with cell culture media. As the reviewer pointed out, cell culture medium contains many supplements that influence the fluid properties. However, the trend observed with the FITC-doped water was similar to that seen with the cell culture media, albeit with different time scales owing

to the influence of fluid properties. We have added both the data (see Figure S1D provided in the response to the previous comment) and a line in the edited manuscript.

Page 6; line 8-10:

We also conducted burst pressure tests with FITC-doped cell culture medium and saw a similar trend albeit with slightly different times owing to different fluid properties (**Figure S1D**).

Comment 3: Page 8, Line 17: *To investigate whether the proposed approach led to the formation of necrotic cores, the authors carried out OCR analysis, which may enable the estimation of reduced metabolism and thereby indirectly assess the presence of necrotic cores. However, there are several direct methods to measure the formation of necrotic cores within engineered tissues, such as live/dead cell analysis. Including results from a live/dead cell analysis would provide a more precise and straightforward assessment of necrotic core formation resulting from the proposed approach.*

Response: We would like to thank the reviewer for the suggestion. We conducted live-dead assay on tissues cultured in static and the lowest flowrate (2 $\mu\text{L/hr}$) and did not see any necrotic core formation in the tissues. We have added a line and data shown for the reviewer.

Page 10; line 10-13:

We further performed live-dead (calcein AM + Ethidium homodimer I) assay on our tissues at static culture (i.e. no flow in the media channel) on day 7, and did not see any necrotic core formation (**Figure S2C**).

Figure S2: (A) Tissue OCR B probability distributions obtained for V1, V2, and V3 using FEM PoM method. **(B)** Differential evolution approach was used to obtain parameters for probability collapse. Show here is the estimation of parameters ν and ζ for the collapse of probability distributions of m to estimate the scaling with respect to L and t , described by a scaling function f . **(C)** Representative images of calcein AM and ethidium homodimer I staining to evaluate live and dead cells within tissues cultured at static and under perfusion of 2 $\mu\text{L/hr}$. Images shown are in the middle of the tissue height.

Comment 4: Page 9, Line 17: *The experiments and comparisons discussed in this section are unclear. This part is crucial as it demonstrates the diffusion capability of the proposed approach for maintaining the physiological functions of engineered microtissues, which is the primary aim of the proposed approach. While measuring twitch amplitude, CaTD80, and beating frequency are robust methods for assessing the function of cardiac microtissues, the comparison between a single perfusion rate and a static culture condition raises concerns regarding the relevance of this experiment to the primary aim. What I understand is that the focus of this experiment is not to examine the effects of shear stress on tissue functions. Additionally, the comparison under static conditions suggests inadequate diffusion of cell culture molecules in the proposed device. Including an additional experimental condition using another diffusion barrier method, such as the microchannel-based approach, would provide a meaningful comparison and help establish the advantages of the proposed method in maintaining in vitro microtissues.*

Alternatively, if the authors present comparisons between varying flow rate conditions, it might be more meaningful as it would allow for an estimation of how the tunability of diffusion, a key advantage of this proposed approach, affects the culture of cardiac microtissues. However, the current comparison seems to primarily investigate how shear stress impacts the culture of microtissues.

Response: The point is well taken. Per the suggestion of the reviewer, we have added another flow condition (2 $\mu\text{L/hr}$) to our dataset in Figure 4. We have chosen this flow condition to show how the tunability of diffusion can play a major role in the functional maintenance of the tissue. This dataset along with the revised Figure 3 should convince the readers about the importance of tuning diffusion rates to achieve functional status of the tissues.

Page 10; line 31-40:

Based on the OCR profiles predicted by our model, the microtissue should not undergo functional decay if there is sufficient diffusion of nutrients into the tissue. **We further performed live-dead (calcein AM + Ethidium homodimer I) assay on our tissues at static culture (i.e. no flow in the media channel) on day 7, and did not see any necrotic core formation (Figure S2C).** However, to test if there were any differences in tissue behavior due to varying diffusion dynamics of nutrients (per Figure 3) ~~To test this, we compared our microtissues cultured in static vs under dynamic medium perfusion at 2 and 20 $\mu\text{L/h}$. This value of 20 $\mu\text{L/h}$ dynamic perfusion rate was chosen so that the shear stress at the walls was ~ 1 dynes/cm², which is considered near physiological.^{47,48} The hiPSC-derived cardiac tissue exhibit automaticity (i.e. spontaneous beating without electrical stimulation).⁴⁹ To test the functionality of the microtissue, we used ~~two~~ ~~three~~ metrics: twitch amplitude, ~~and~~ calcium transient duration at 80% repolarization percentage (CaTD₈₀), ~~and beating frequency.~~ Twitch amplitude is associated with the contractile nature of the cardiac tissue, i.e. how much the tissue contracts during a spontaneous beat. The CaTD₈₀ provides a proxy measurement of the membrane potential waveform of the cardiac tissue ~~and the beating rate provides the rate of spontaneous beating.~~⁴⁹ We used published methods to measure the twitch amplitude, ~~and CaTD80, and the equivalent beating rate~~ of the microtissues.^{3,23,50} We monitored the ~~spontaneous~~ beating of the hiPSC-derived cardiac tissue~~

over 1 week/7 days. Microtissues cultured in static condition were fed every 2 days with fresh medium. Across different batches of measurements, 2/3rd of the microtissues cultured in static conditions failed to beat ~~spontaneously~~, indicative of functional ~~alteration-degradation~~ (Figure 4G). In contrast, 92% of the microtissues in dynamic perfusion at 20 $\mu\text{L/h}$ had ~~spontaneous~~-beating, indicative of the functional integrity over 7 days. ~~The fraction of spontaneously beating tissues cultured with 2 $\mu\text{L/h}$ was approximately 67%. We hypothesize that the varying fraction of beating tissues across these conditions were primarily driven by diffusion dynamics and advective current of biomolecules across the barrier interface as predicted by Figure 3B.~~ These microtissues with ~~spontaneous~~-beating at day 7 were considered functionally active tissues. Within these functionally active tissues, static culture led to the peak twitch amplitude of the microtissue being slightly reduced on day 7 (Figure 4H), whereas there was no statistical difference between the microtissues on day 0 vs day 7 in MPS where there was dynamic perfusion. The microtissues cultured in flow had higher CaTD80 ~~and beating rate, albeit not statistically significant and beating rate compared to static cultured microtissues~~ (Figure 4 I). Thus, the pillar array allows the diffusion of nutrients and metabolites to maintain the functionality of physiologically dense 3D cardiac tissue.

Figure 4: Pillar array allows the generation physiologically relevant microtissue. (A) 3D cardiac tissue formed within the MPS after aggregation of the single-cell. Shown are the brightfield image, and confocal images of the microtissue stained with nuclear stain DRAQ5. **(B)** Quantification of the cell number, volume, and density within the MPS. **(C)** Trace of mitochondrial oxygen consumption rates of cardiomyocytes in 2D obtained via Seahorse respirometer ($n=10$; error bars represent SEM). **(D)** Computational domain used for the FEM PoM to predict oxygen profiles within the 3D tissue, and distribution of sOCR utilized in the reaction term. **(E)** Representative image of 3D oxygen profile obtained using the PoM. Shown here is the geometry V2. **(F)** Log-log distribution of collapse of the tissue OCR to determine the relationship between tissue OCR B and mass of microtissue m . $\gamma = 0.125$ and $\alpha = 0.8825-0.95$, as determined by finite size scaling. **(G)** Comparison of percentage tissues beating after day 7 in static vs. flow conditions. **(H)** Twitch amplitude comparison between day 0 vs day 7. Significance between day 0 and day 7 tested using student's t-test. **(I)** Comparison of calcium transient duration CaTD_{80} of microtissues on day 7 for static vs flow conditions.

Comment 5: *page 11, line 34. 'We note that this is similar to any membrane-based barrier, where endothelial cells only cover the porous barrier [8, 46].' I am not sure whether this discussion is correct. What I have known about the two citations, the four surfaces of their vascular channels separated by porous membranes from other channels, are fully lined with endothelial cells. (in reference 8, figure 1C and in reference 46, figure 1a). Please look into these and provide correct discussions.*

Response: We concur that the statement was confusing, and therefore we have removed it from our manuscript.

Comment 6: *Page 11, Line 35: I am wondering if this discussion is also correct. The cellular merging observed in the area of the pillar array seems to result from the migration of endothelial cells toward the center culture chamber, rather than from intercellular interactions between endothelial cells from the media channel and cardiac myocytes from the center chamber.*

Response: We thank the reviewer for raising this concern about the clarity of the statement. We have rewritten the sentences to make sure our statements were not confusing, and the intent of the experiment was clear.

Page 12; line 29-36:

We performed confocal imaging to assess **how well** the HCAECs **coverage** within the channel (as envisioned in Figure 5A) **and whether the endothelial cells covered the pillar barrier openings in the z direction** to add an active element to the passive barrier. We observed that the HCAECs covered the fenestrations, but did not form a 3D lumen covering all 4 walls of the channel (Figure 5G). ~~We note that this is similar to any membrane-based barrier, where endothelial cells only cover the porous barrier.^{8,46} It is also noted that our pillar array diffusion barrier allows the formation of contiguous cell-cell interaction at the interface (Figure 5F), rather than discrete nature of contact using microchannel-based fenestrations.¹⁴~~ The pillar barrier allows the endothelial cells to form a relatively contiguous interface (Figure 5F) that covers the pillar barrier openings in the z direction allowing the formation of an active component of the barrier.

Comment 7: *Page 11, Line 38: The migration of endothelial cells into the pillar array area may limit the diffusion of culture media molecules between the center and media chambers. Thus, to demonstrate the co-culture capability of the proposed approach, specifically for co-culturing endothelial cells in the media channel with cardiac myocytes and fibroblasts in the center chamber, it is necessary to investigate the viability, twitch amplitude, CaTD80, and beating frequency of the cardiac microtissues over 7 days, similar to what was done in the previous section.*

Response: We intend this manuscript to focus on the engineering of the pillar interface barrier. While the reviewer's suggested experiments are interesting, we believe these will be beyond the scope of this paper, as things such as scalability between cells/tissues and common medium could all impact the nature of the outcomes. We intend on pursuing these in future studies. We have added lines in our modified manuscript clearly stating the limitations of our present study.

Page 12 line 46 – Page 13 line 3:

In vitro cardiac MPS consisting of heterotypic cellular components such as endothelial cells and macrophages have been proposed to create more physiologically relevant organ models and remains the scope of future investigation leveraging our MPS platform with tunable barrier interface.

Page 13; line 12-19:

In this study we have demonstrated a highly tunable pillar interfacial barrier for MPS that can be leveraged for multi-tissue/cellular models. We haven't addressed issues of scaling of tissues/cells with respect to each other, or design of a common medium required for functional coupling of cells/tissues. These remain challenges within literature, and scope for future study. Finally, while we have shown the ease of fabrication of these pillar interfacial barriers with PDMS and replica molding, other fabrication strategies such as embossing/thermoforming with materials like cyclic olefin or polycarbonate will need future investigations.

Comment 8 (minor): *Page 2, Line 13: 'which results in filling of the tissue chambers' is confusing. It seems it needs to be corrected to 'which results in filling of the media channels.'*

Response: We have removed that part of the sentence to avoid confusion. Cells are loaded into the tissue chambers via centrifugation or application of negative pressure.

Page 2; line 17-19:

Furthermore, a common procedure to load cells into the microfluidic tissue chambers involves the centrifugation of cells into the chamber or application of a vacuum, ~~which results in filling of the tissue chambers.~~

Comment 9 (minor): *In Figure 5F: Do red and green indicate VE-cadherin and CD31, respectively? Although Figure 5E provides indicators, it would be clearer if a dotted line extended from Figure 5E to Figure 5F, similar to what was done in Figure 5H.*

Response: Thank you for pointing out this. We have fixed it in the revised version of our manuscript. (Please refer to page 2 of this document).

Comment 10 (minor): *In Figure 5G: It is unclear where the media channel and pillar array area are located in this figure. Why are endothelial cells shown on the 10 μm surface and not on the 0 μm surface? This figure seems to label the 26 μm for the top-surface of the media (vascular) channel, but why are there no endothelial cells there? This observation is related to the comment #5.*

Response: The endothelial cells are in the media channel, which was clarified in the original document (see lines below from the original text). We performed confocal imaging to see if the HCAECs covered the pillar array openings (at 10 μm). The endothelial cells are on both the 0 and 10 μm surface, as seen by the nuclear stain. A faint endothelial signal were detected at the bottom of the layer, reaching peak expression at a depth of 10 μm . It wasn't detected beyond 26 μm . Detailed response was presented to Comment 6.

Page 12; line 23-24:

We cultured human coronary artery endothelial cells (HCAECs) in the media channels of the cardiac MPS.

Reviewer 3

Comment 1: *The idea of generating microfluidic devices with tunable porosity is potentially interesting. However I see a major drawback related to the pore size of 8 μm , which looks to high for any type of biological barrier (standard membranes have pores around 0.45 μm). Indeed, as shown also by author the cells go through the barrier which is actually not really physiological. It would also be difficult to monitor barrier integrity with standard method such as TEER as well to do any significant evaluation regarding the passage of substances. The author should justify better these issues.*

Response: Pore sizes larger than 0.45 μm are regularly used in microphysiological systems, ranging from 2 μm (e.g. Mathur et al. Sci. Rep. 2015) to 10 μm (Huh et al. Science 2010). Furthermore, we have shown the integration of impedance measurement in MPS in our recent study (Kim et al. Small 2025).

We would like to thank the reviewers for their time to thoroughly review our paper. We thank the reviewers for acknowledging our efforts to address each comment with the outmost rigor. Hence, our current manuscript is the final version with formatting changes per the journal.

Reviewer 2

Comment 1: *The authors addressed all of the concerns I raised by revising the manuscript, conducting additional experiments, or acknowledging them as limitations of the proposed approach.*

Response: We thank the reviewer for acknowledging our efforts in addressing all concerns.

Reviewer 3

Comment 1: *After revision the authors significantly improved the impact of the manuscript. I just suggest to include a discussion on the use of different pore size which is crucial for this study.*

Response: Figure 3 and associated discussion addresses the reviewer's comment.